# Subventricular zone cytogenesis provides trophic support for neural repair in a mouse model of stroke

Michael R. Williamson [1,2] ✉, Stephanie P. Le[3], Ronald L. Franzen [3,4], Nicole A. Donlan[3], Jill L. Rosow[3], Mathilda S. Nicot-Cartsonis[5], Alexis Cervantes[2,6], Benjamin Deneen [2,6], Andrew K. Dunn[1,7], Theresa A. Jones [1,3] & Michael R. Drew [1,8]

Stroke enhances proliferation of neural precursor cells within the subventricular zone (SVZ) and induces ectopic migration of newborn cells towards the site of injury. Here, we characterize the identity of cells arising from the SVZ after stroke and uncover a mechanism through which they facilitate neural repair and functional recovery. With genetic lineage tracing, we show that SVZ-derived cells that migrate towards cortical photothrombotic stroke in mice are predominantly undifferentiated precursors. We find that ablation of neural precursor cells or conditional knockout of VEGF impairs neuronal and vascular reparative responses and worsens recovery. Replacement of VEGF is sufficient to induce neural repair and recovery. We also provide evidence that CXCL12 from peri-infarct vasculature signals to CXCR4-expressing cells arising from the SVZ to direct their ectopic migration. These results support a model in which vasculature surrounding the site of injury attracts cells from the SVZ, and these cells subsequently provide trophic support that drives neural repair and recovery.

Functional recovery is often limited after damage to the central nervous system. Consequently, stroke and other forms of brain injury often cause long-lasting disabilities. Following stroke, remodeling of residual tissue surrounding the site of injury is thought to underlie recovery. For example, extensive plasticity of neural circuits and blood vessels occurs in peri-infarct regions and these processes are associated with functional improvement[1–4] (cf[5].). Repair processes are mediated by interactions across disparate cell types[1,2,6–8]. However, the intercellular interactions that orchestrate repair and recovery remain to be completely defined. An improved understanding of the mechanisms that govern neural repair could inform development of new treatment strategies.

Cytogenesis, the formation of new cells, is limited in the adult mammalian brain. The subventricular zone (SVZ) is one of a small number of regions that contains multipotent neural stem and progenitor cells (collectively referred to here as precursors) that generate new neurons and glia in adulthood[9,10]. Normally, the predominant progeny arising from the SVZ are new neurons that migrate towards the olfactory bulb and integrate into existing olfactory circuitry. However, injuries such as stroke markedly increase SVZ precursor proliferation and induce ectopic migration of SVZ-derived cells towards the site of injury[11–16]. Past studies of this process after brain injury have largely focused on neurogenesis − the formation of new neurons and their localization in peri-infarct regions[13,14]. In general,

[1]Institute for Neuroscience, University of Texas at Austin, Austin, TX, USA. [2]Center for Cell and Gene Therapy, Baylor College of Medicine, Houston, TX, USA. [3]Department of Psychology, University of Texas at Austin, Austin, TX, USA. [4]School of Medicine, Baylor College of Medicine, Houston, TX, USA. [5]John Sealy School of Medicine, University of Texas Medical Branch at Galveston, Galveston, TX, USA. [6]Center for Cancer Neuroscience and Department of Neurosurgery, Baylor College of Medicine, Houston, TX, USA. [7]Department of Biomedical Engineering, University of Texas at Austin, Austin, TX, USA. [8]Center for Learning and Memory and Department of Neuroscience, University of Texas at Austin, Austin, TX, USA. ✉e-mail: mrwillia@utexas.edu

experimental manipulations that increase post-stroke neurogenesis are associated with enhanced functional recovery[15]. A common view has been that cell replacement, especially neuron replacement, could allow for partial brain regeneration and consequently improved function. However, recent findings indicate that new neurons poorly integrate into existing circuits in peri-infarct regions and receive little synaptic input[17], making their functional importance unclear (cf[18].). Moreover, other studies have found that astrocytes outnumber neurons among migrating SVZ-derived cells, but the entire population of SVZ-derived cells has yet to be comprehensively characterized[11,19,20]. Overall, the identity and functional importance of new cells that arise from the SVZ after injury are not well understood.

Here we investigate the SVZ cytogenic response following cortical photothrombotic stroke in mice. Our goals were to characterize the types of cells produced by the SVZ after stroke and mechanistically understand the role of SVZ cytogenesis in stroke recovery. We use indelible lineage tracing to phenotype SVZ-derived cells that migrate towards the site of injury. Unexpectedly, we find that the majority of these cells are undifferentiated precursors. The ectopic migration of cells from the SVZ towards the lesion is directed, at least in part, via CXCL12-CXCR4 signaling. Reducing cytogenesis impairs motor recovery after stroke, at least in part due to deficits in neuronal and vascular plasticity. With gain- and loss-of-function manipulations, we show that VEGF produced by SVZ-derived cells drives repair and functional recovery. These findings identify SVZ cytogenesis as a source of trophic support that facilitates neural repair. Thus, our study demonstrates a mechanism by which endogenous neural precursor cells contribute to repair and recovery in the injured central nervous system.

## Results

### Cells arising from the subventricular zone after stroke are predominantly quiescent precursors and astrocytes

We used genetic lineage tracing to characterize the SVZ cytogenic response to stroke. Young adult (3–6 months old) *Nestin*[CreERT2]; Ai14 mice were injected with tamoxifen to induce indelible tdTomato expression in neural stem cells and their progeny (Fig. 1a, b)[11,19]. Four weeks later, photothrombotic cortical infarcts were induced, and tissue was collected one, two, six, or eight weeks post-stroke. The first two time points correspond to a period when substantial neural repair is ongoing and functional improvement is incomplete[7,8,17]. The latter two time points correspond to times after which recovery is largely complete. While no cortical migration was seen in the absence of injury, unilateral cortical stroke induced a profound migration of tdTomato+ cells from the SVZ into peri-infarct cortex (Fig. 1c, d). We immunostained tissue to examine expression of an array of differentiation stage-specific and proliferation-associated proteins in lineage-traced cells (Fig. 1e–s, Supplementary Figs. 1, 2).

Unexpectedly, the majority of tdTomato+ cells in peri-infarct cortex expressed precursor cell-associated markers (>90% were CD133+ / Sox2+, and ~65% were Ascl1+). ~25–30% of tdTomato+ cells were differentiated astrocytes based on expression of S100β, which has been shown to define astrocyte maturation and loss of multipotency[21,22] and is not expressed by SVZ precursors[23]. Astrocyte reactivity is associated with re-expression of some precursor cell-associated proteins, including CD133 and Sox2[24,25], but reactive astrocytes do not express Ascl1[26,27]. Thus, the expression of Ascl1 defined undifferentiated precursors, S100β expression defined astrocytes, and CD133/Sox2 labeled both subpopulations (Fig. 1e). Lineage-traced cells were largely quiescent as defined by expression of the quiescence marker[28] Id2 in >90% of cells, and rare expression of the proliferation marker Ki67 (<5%; Fig. 1q). Oligodendrocyte-lineage (Olig2+, < 5%) and neuron-lineage cells (DCX+, < 2%, and NeuN+, < 1%) made up the remainder of lineage-traced cells. We corroborated these phenotyping results with parallel experiments in *Ascl1*[CreERT2]; Ai14 mice, in which a subset of neural stem and progenitor cells are lineage-traced[18] (Supplementary Fig. 1). The

distribution of cell identities was similar across all time points examined (Fig. 1r, s), indicating that there is no discernable differentiation by the time behavioral recovery is complete. We also confirmed that few new neurons are formed up to 6 weeks post-stroke using *Nestin*[CreERT2]; *Rosa26*[CAG-LSL-Sun1-sfGFP] mice in which neural stem cells and their progeny were labeled with a nuclear membrane-targeted fluorophore (Supplementary Fig. 2). There was a progressive increase in the number of lineage traced cells in peri-infarct cortex from 1 to 6 weeks after stroke, but no further increase afterwards, suggesting a continuous production of these cells during the recovery period (Fig. 1p). Given the rare expression of Ki67 in SVZ-derived cells localized to cortex (Fig. 1q), it is most likely that the increase in cell numbers over time is mainly driven by migration from the SVZ. These experiments identify undifferentiated precursors as the predominant cell type produced by the SVZ in response to photothrombotic stroke.

### Chemogenetic ablation of neural stem cells impairs recovery after stroke

Having characterized the phenotype of SVZ-derived cells, we next investigated whether SVZ cytogenesis is beneficial for recovery after stroke. We used *GFAP-TK* mice to selectively ablate neural stem cells prior to stroke following an established ganciclovir (GCV) administration paradigm[9,29] (Fig. 2a–j). Importantly, GFAP-TK mice allow for ablation of the same population targeted by the *Nestin*[CreERT2] mice we used for lineage tracing[9,10,30,31] (Supplementary Fig. 3). Delivery of GCV via subcutaneous osmotic pumps for two weeks ablates mitotic TK-expressing cells (i.e. neural stem cells), but spares non-mitotic GFAP-expressing cells, including cortical astrocytes[9,29] (Fig. 2d–f; Supplementary Fig. 4). SVZ cytogenesis, as measured by the density of SVZ DCX+ and Ki67+ cells, was substantially reduced in *GFAP-TK* + GCV mice relative to littermate controls, which included wildtype mice given saline or GCV and *GFAP-TK* mice given saline (Fig. 2g–j).

Following GCV administration, we compared motor recovery between *GFAP-TK* mice and wildtype littermates after photothrombotic cortical infarcts targeting the forelimb area of motor cortex[7,32] (Fig. 2k). Motor function was assessed with the single seed reaching task, a highly sensitive and translationally relevant measure of skilled reaching[33,34]. There was no difference between groups in reaching performance during pre-stroke GCV delivery ($p \geq 0.84$, Tukey tests). Mice lacking cytogenesis showed significantly worse recovery out to four weeks following stroke (Fig. 2l). We also examined control groups consisting of wildtype mice given saline and GFAP-TK mice given saline, which showed similar recovery with wildtype mice given GCV (Fig. 2l). Lesion size and location did not differ between groups (Fig. 2m, n). These results demonstrate that SVZ cytogenesis promotes functional recovery after stroke.

### Aging diminishes the SVZ cytogenic response to stroke and its functional benefits

Aging is associated with a substantially higher incidence of stroke[35] and reduced recovery of function in humans[36]. In animals, aging is associated with diminished SVZ cytogenesis[37–40]. We examined the impact of aging on normal and post-stroke SVZ cytogenesis by comparing young adult (aged 3–6 months) and aged (12–16 months) *Nestin*[CreERT2]; Ai14 mice. Mice were either uninjured (naïve) or subjected to cortical stroke two weeks prior to tissue collection. In young mice, stroke increased the number of Ki67+ proliferative cells and tdTomato+Sox2+ precursors in the SVZ relative to young naïve, aged naïve, and aged stroke mice (Fig. 3a–d). By contrast, in aged mice, there was no significant difference in the number of SVZ Ki67+ or tdTomato+Sox2+ cells after stroke relative to naïve mice ($p = 0.604$, $p = 0.998$, respectively, Tukey tests). Thus, stroke increases SVZ proliferation and expands the precursor cell pool in young, but not aged, mice.

We next examined the effects of aging on the migratory response of lineage-traced SVZ cells after stroke. The density of

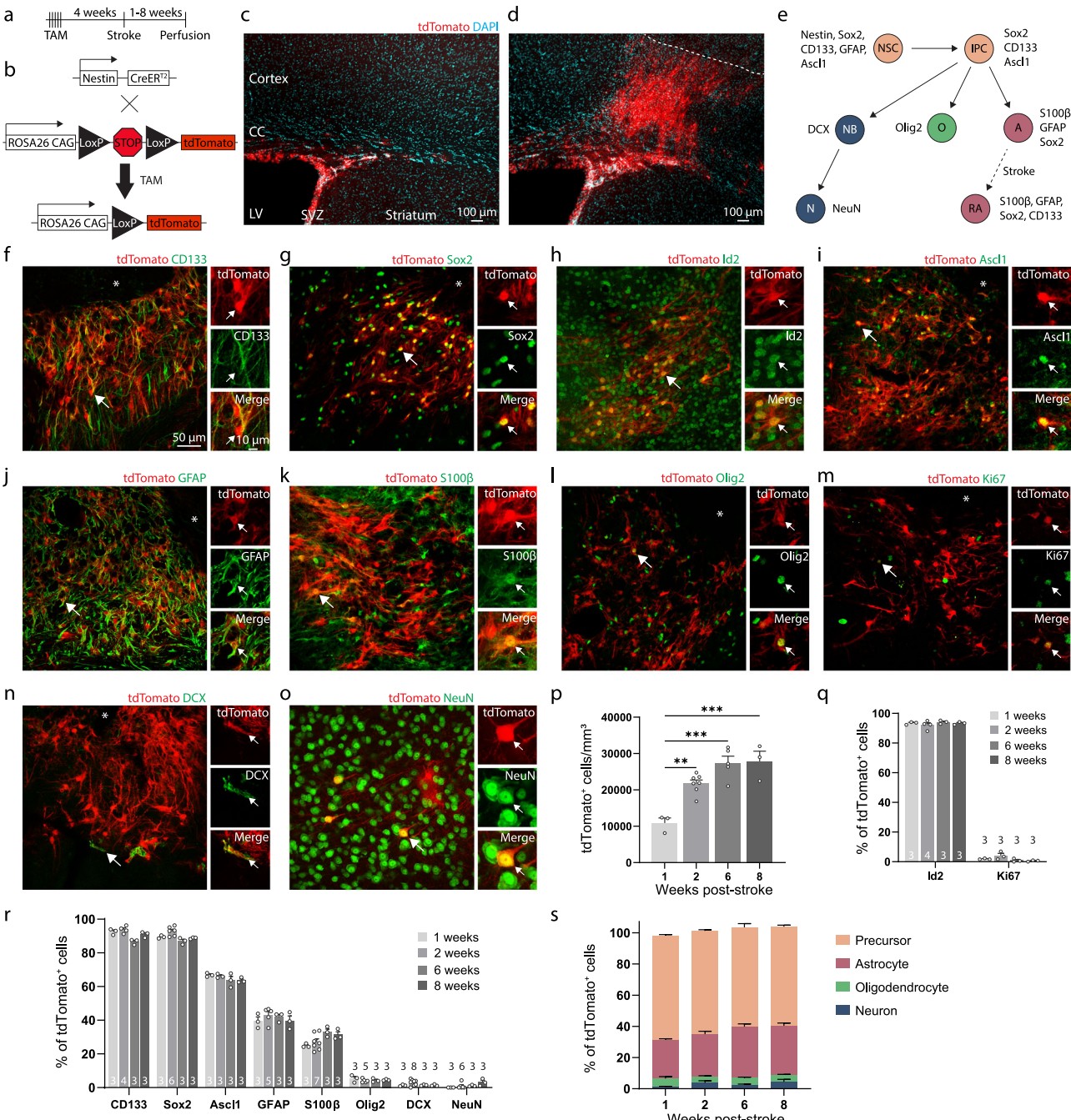

**Fig. 1 | SVZ-derived cells are predominantly undifferentiated precursors.**
**a** Experimental timeline. **b** Schematic of lineage tracing system. **c** Image of tdTomato-expressing cells in the subventricular zone in the absence of injury. CC, corpus callosum; LV, lateral ventricle; SVZ, subventricular zone. **d** Substantial migration of tdTomato⁺ cells towards the infarct after cortical photothrombotic stroke (dashed line indicates the approximate infarct border). Images in (**c**, **d**) are representative of experiments repeated in 4 mice. **e** Schematic of differentiation stages as defined by marker expression. Neural stem cells (NSC) produce intermediate progenitor cells (IPC), which give rise to cells of the three major neural lineages: neurons (NB, neuroblast; N, neuron), oligodendrocytes (O), and astrocytes (A). After stroke, reactive astrocytes (RA) re-express some neural stem cell markers. **f–o** Representative confocal images from peri-infarct cortex of tdTomato⁺ cells and immunostaining for various markers. Asterisks indicate the lesion core. Scale is the same for (**f–o**). Staining for: (**f**) CD133, (**g**) Sox2, (**h**) Id2, (**i**) Ascl1, (**j**)

GFAP, (**k**) S100β, (**l**) Olig2, (**m**) Ki67, (**n**) NeuN, and (**o**) DCX. **p** Quantification of the number of tdTomato⁺ cells in peri-infarct cortex across time after stroke. **p** = 0.0025, ***$p \leq 0.0002$, one-way ANOVA and Tukey's post-hoc tests. $n = 3$ (1 week), 7 (2 week), 5 (6 week), 3 (8 week) mice. **q** Quantification of quiescence- (Id2) and proliferation-related (Ki67) marker expression. The number of mice used for each marker is indicated on the bottom of each bar. **r** Quantification of differentiation stage and lineage-specific marker expression in tdTomato⁺ cells. The number of mice used for each marker is indicated on the bottom of each bar. **s** Estimate of phenotype distribution of lineage traced cells. n is the same as in panel **r**. Precursors were defined by Ascl1. Astrocytes were defined by S100β. Oligodendrocyte lineage was defined by Olig2. Neuron lineage was defined by DCX and NeuN. Data are presented as mean ± SEM. Source data are provided as a Source Data file.

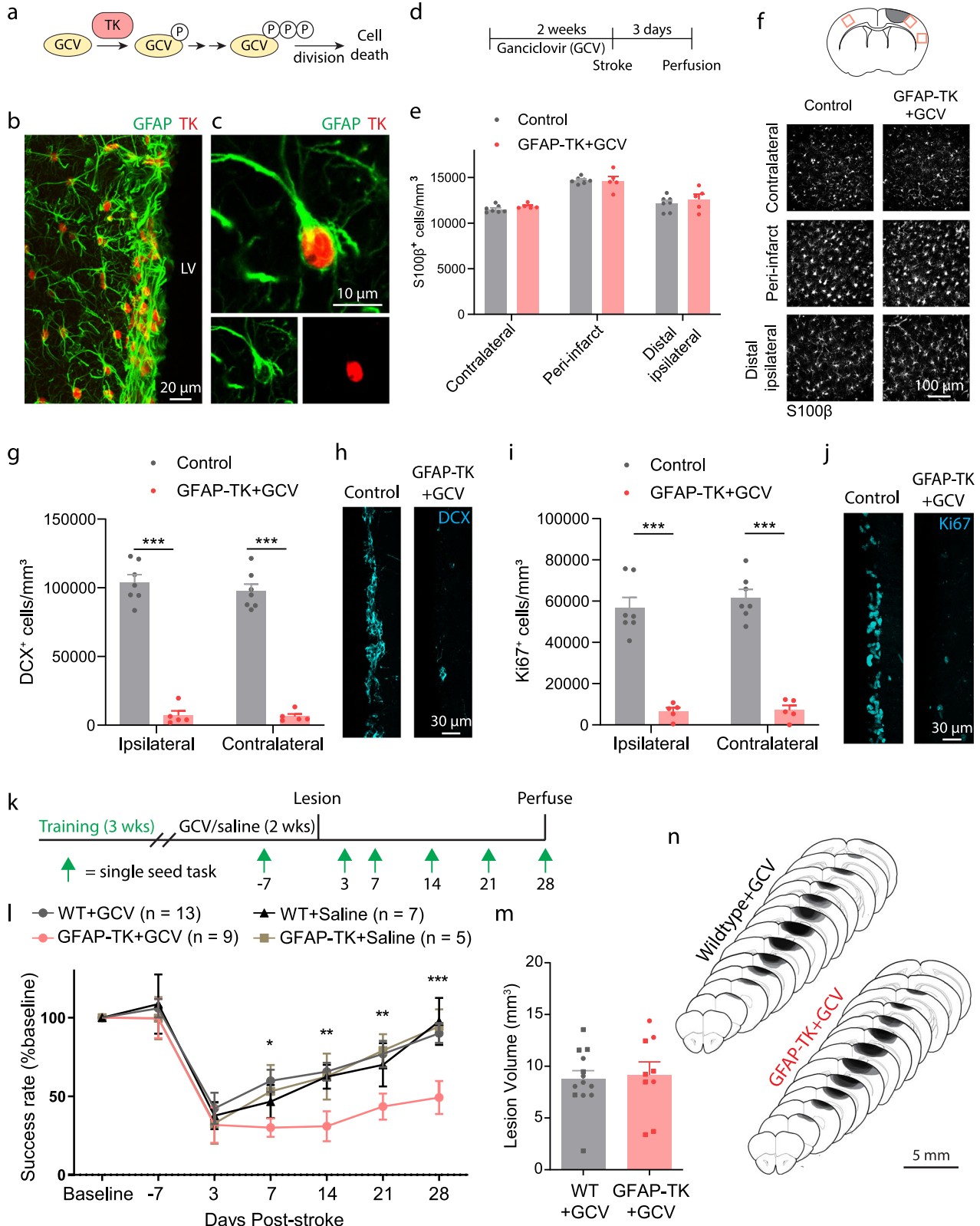

tdTomato[+] cells in peri-infarct cortex two weeks post-stroke was about five-fold lower in aged mice relative to young mice (Fig. 3e, f). Despite the diminished cytogenic response, the phenotype distribution of tdTomato[+] cells in peri-infarct cortex of aged mice (Fig. 3g) was similar to what we observed in young adult mice (Fig. 1), with the majority of cells identified as precursors and astrocytes. Thus, far fewer SVZ-derived cells localize to peri-infarct

regions in aged mice, but their phenotype distribution is similar to that of young mice.

We next investigated whether the blunted cytogenic response observed in aged mice was still functionally beneficial (Fig. 3h). We used *GFAP-TK* mice to selectively ablate neural stem cells prior to stroke. BrdU pulse labeling of proliferating cells revealed a near complete loss of BrdU[+] cells in the SVZ of *GFAP-TK* + GCV mice relative

**Fig. 2 | Ablation of neural stem cells worsens recovery after stroke. a** Schematic illustrating the thymidine kinase (TK)/ganciclovir (GCV) system for conditional ablation of proliferating cells. **b** Confocal image showing TK expression in GFAP⁺ cells of the subventricular zone in a GFAP-TK mouse. LV, lateral ventricle. **c** High-resolution image of a TK-expressing SVZ stem cell. Images in (**b, c**) are representative of experiments repeated in 3 mice. **d** Experimental timeline for assessing the specificity and effectiveness of arresting SVZ cytogenesis with GFAP-TK mice (**E–J**). GCV or saline was delivered for 14 days via osmotic pump. Tissue was collected 3 days after stroke. **e** Parenchymal astrocytes were not depleted in this paradigm. t(10) < 1.53, $p \geq 0.157$, two-tailed $t$ tests comparing groups for each region. $n = 7$ control (wildtype mice given GCV or GFAP-TK mice given saline), $n = 5$ GFAP-TK + GCV mice. **f** Representative images of S100β⁺ cells from three cortical regions (see diagram at top) show lack of astrocyte ablation. **g–j** GFAP-TK mice permit conditional arrest of cytogenesis. The number of SVZ DCX⁺ (**g, h**) and Ki67⁺

cells (**i, j**) was significantly reduced in GFAP-TK mice given GCV ($n = 5$) relative to controls ($n = 7$). \*\*\*t(10) ≥ 7.99, $p < 0.0001$, two-way $t$ tests. **k** Experimental design for assessing motor recovery after photothrombotic stroke with the single seed reaching task. **l** Arrest of cytogenesis significantly worsened recovery of motor function. Time x group interaction $F_{(18, 174)} = 2.19$, $p = 0.0052$. \*$p = 0.030$, \*\*$p \leq 0.007$, \*\*\*$p = 0.0007$ GFAP-TK + GCV compared to at least one control group, two-way ANOVA and post-hoc Tukey's tests. WT = wildtype. **m** Lesion volume was not different between groups. t(20) = 0.27, $p = 0.787$. Two-tailed $t$ test. $n = 13$ WT + GCV, $n = 9$ GFAP-TK + GCV mice. **n** Lesion reconstruction. Darker shades represent more overlap between animals. Data are presented as mean ± SEM. Where individual datapoints are shown, datapoints representing males are shown as circles; datapoints representing females are shown as squares. Source data are provided as a Source Data file.

---

to control aged mice (Fig. 3i, j; all mice aged 12–16 months). We assessed motor recovery with the single-seed task and found no differences between groups (Fig. 3k). Notably, both aged groups performed significantly worse on day 28 than young mice within intact cytogenesis (wildtype+GCV data from Fig. 2; F(3, 35) = 7.46, $p < 0.001$ one-way ANOVA; p < 0.012, Tukey tests), but similarly to young GFAP-TK + GCV mice (p > 0.978, Tukey tests). Lesion size was not different between aged groups (Fig. 3l, m), and was similar to that of young adults (Fig. 2m, n). Altogether, our results indicate that aging diminishes the cytogenic response to stroke and its functional benefits. Together with our findings in young mice, our results show that reduced SVZ cytogenesis, by neural stem cell ablation or aging, is associated with worse functional recovery. Reduced cytogenesis may contribute to worse outcome after stroke with aging.

## Arrest of cytogenesis disrupts neuronal and vascular repair

Our results indicate that most SVZ-derived cells are undifferentiated precursors and reactive astrocytes. We hypothesized that these cell types may influence repair processes to promote behavioral improvement. In particular, synaptic plasticity and vascular remodeling are two functionally important aspects of neural repair that could potentially be augmented by factors produced by SVZ-derived cells. Indeed, several studies have demonstrated beneficial effects of transplanted stem cells and resident cortical astrocytes on these processes[7,41–50].

We investigated the consequences of neural stem cell ablation on synaptic and vascular plasticity in residual cortex surrounding photothrombotic infarcts with a longitudinal imaging approach (Fig. 4a, b). We bred GFAP-TK mice with Thy1-GFP mice, which have sparse, GFP-labeled pyramidal neurons. Resulting double transgenic mice allowed us to monitor synaptic remodeling at single-synapse resolution with repeated 2-photon imaging of dendritic spines on apical dendrites before and after stroke[1,2,51] with the ability to conditionally arrest cytogenesis. Blood flow was tracked with multi-exposure speckle imaging (MESI), a quantitative, optical, contrast-free technique that yields high-resolution blood flow maps[3,52]. Prior to stroke, control and TK⁺/⁻ mice were unilaterally implanted with cranial windows, trained on the single-seed reaching task, administered GCV or saline, and subjected to baseline imaging. We then induced strokes in forelimb motor cortex and periodically imaged dendritic spines and blood flow and assessed behavioral performance during recovery.

Motor recovery was again significantly impaired in GFAP-TK + GCV mice, with no differences in lesion size observed between groups (Fig. 4c–e). 2-photon imaging of dendritic spines revealed increased spine turnover in peri-infarct cortex (Fig. 4f–h), consistent with past work[1,2,6,51]. There were no group differences in spine dynamics before stroke (Supplementary Fig. 5). New spine formation peaked during the second week after stroke, and was significantly higher in mice with intact cytogenesis. Spine elimination was greatest during the first week post-stroke, and subsequently declined with

time, without significant differences between groups. A subpopulation of new spines formed after stroke persists long-term, and the persistence of newly formed spines is associated with greater functional recovery[2]. The survival of new spines was significantly reduced in GFAP-TK + GCV mice regardless of the day of spine formation (Fig. 4i). Moreover, spine survival rate was positively correlated with behavioral performance on the final assessment day (Fig. 4j). Thus, SVZ cytogenesis supports synaptic remodeling after stroke, particularly by promoting the long-term stabilization of new synapses.

Broad regions of reduced blood flow persist for days to weeks surrounding focal strokes[3,52]. Remodeling of peri-infarct vasculature helps to restore blood flow and is associated with behavioral improvement[3]. To examine vascular remodeling, we injected mice with fluorophore-conjugated tomato lectin immediately before euthanasia to label perfused vasculature. Vessel density in peri-infarct cortex was significantly reduced in GFAP-TK + GCV mice relative to controls (Fig. 4k, l). GCV administration in wildtype did not affect vascular density, which indicates that the reduction in vascular density in GFAP-TK + GCV mice is not attributable to GCV itself (Supplementary Fig. 6). Furthermore, longitudinal blood flow imaging demonstrated impaired recovery of blood flow at days 5 and 28 post-stroke (Fig. 4m, n). Early vascular permeability was not affected by arresting cytogenesis (Supplementary Fig. 4). Overall, these findings demonstrate that SVZ-derived cells beneficially shape neuronal and vascular remodeling processes after stroke in order to promote recovery.

## SVZ-derived cells interact with vasculature and produce trophic factors

Interactions between blood vessels and neuroblasts migrating towards peri-infarct regions have been previously described[12,53]. We observed frequent contact between lineage-traced SVZ progeny and blood vessels in peri-infarct cortex, including contact of nearby vessels by the processes or cell bodies of 88.2% of SVZ-derived cells (Fig. 5a–c). Thus, interactions with blood vessels may underlie some of the reparative effects of SVZ-derived cells. In addition, stroke may induce expression of migratory cues in endothelial cells to drive migration of reparative cells from the SVZ[12].

Transplanted neural precursors of various sources and in diverse disease settings have been reported to express trophic factors, which may be implicated in their therapeutic effects and may be independent of cell replacement[42,43,46–48,54–56]. To prospectively investigate molecular mechanisms underlying the facilitation of post-stroke repair by SVZ cytogenesis, we examined the expression of trophic factors known to drive neuronal and vascular growth. ~90% of all SVZ-derived lineage traced cells in peri-infarct cortex expressed VEGF, BDNF, GDNF, and FGF2 (Fig. 5d–h). We compared relative abundance of each of these proteins in peri-infarct cortex between control and GFAP-TK + GCV mice at 28 days post-stroke to evaluate production of these factors by SVZ-derived cells relative to other cell types. VEGF protein was

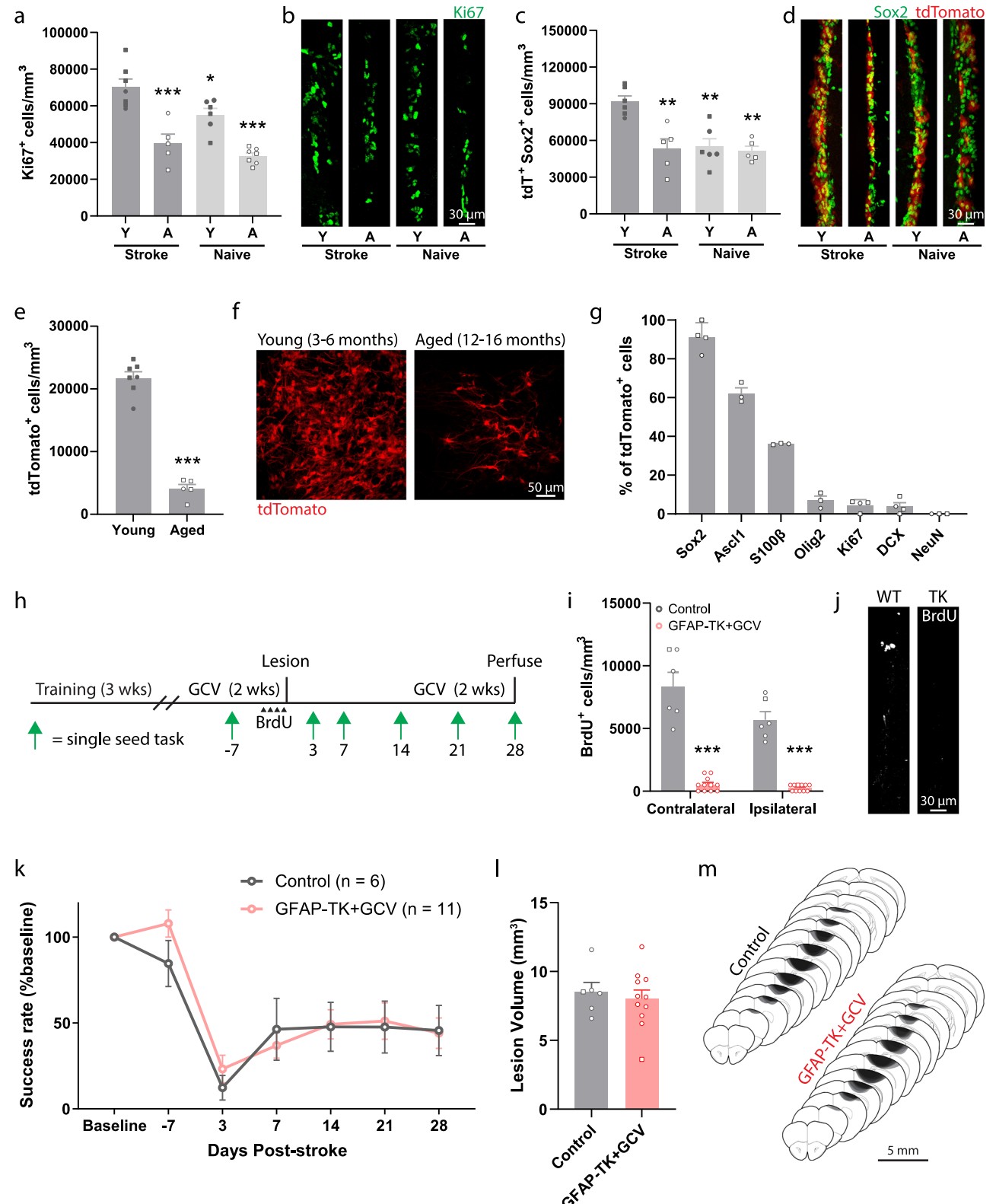

significantly reduced in *GFAP-TK* + GCV mice, indicating that SVZ-derived cells are a major source of VEGF (Fig. 5i). Notably, among SVZ-derived cells, all lineages except for new neurons produced VEGF (Fig. 5j). These findings suggest that SVZ cytogenesis may facilitate neural repair at least in part through production of trophic factors, particularly VEGF. Based on these findings, we focused on VEGF as a potential mechanism by which SVZ-derived cells promote recovery in subsequent experiments.

## Conditional deletion of Vegf in adult neural stem cells impairs recovery and repair

VEGF promotes the growth of blood vessels and neurons[57-60]. Since VEGF was uniquely highly expressed in SVZ-derived cells relative to cortical cells, we next investigated whether VEGF produced by cells arising from the SVZ was involved in post-stroke recovery and repair. We generated *Nestin*^*CreERT2*; *Vegf*^fl/fl mice to permit inducible deletion of *Vegf* in adult neural stem cells and their progeny (VEGF cKO; Fig. 6a).

**Fig. 3 | Aging diminishes the cytogenic response to stroke and its contribution to recovery. a–d** Assessment of SVZ cytogenesis in young (Y, 3–6 months) and aged (A, 12–16 months) *Nestin^CreERT2*; Ai14 mice. Tissue was collected two weeks post-stroke. The number of proliferating cells (Ki67⁺, (**a**, **b**); *n* = 7 young+stroke, 5 aged +stroke, 6 young+naïve, 7 aged+naïve mice.) and neural precursor cells (tdTomato⁺ Sox2⁺, (**c**, **d**); *n* = 6 young+stroke, 5 aged+stroke, 6 young+naïve, 5 aged+naïve mice.) in the SVZ was significantly higher in young mice after stroke compared to all other groups. **p* = 0.036, ***p* ≤ 0.0016, ****p* < 0.0001 relative to young stroke mice, one-way ANOVA and post-hoc Tukey tests. **e** Fewer tdTomato⁺ cells were observed in peri-infarct cortex of aged mice two weeks post-stroke. ****t*(10) = 13.4, *p* < 0.0001. Two-tailed *t* test. *n* = 7 young, 5 aged mice. **f** Representative images of tdTomato⁺ cells in peri-infarct cortex. **g** Quantification of lineage marker expression by tdTomato⁺ cells in peri-infarct cortex of aged mice at two weeks post-stroke (*n* = 3 or 4 mice per marker). Similar to what was observed in young mice (Fig. 1), most

SVZ-derived cells were undifferentiated precursors or astrocytes. **h** Experimental timeline for examining the effects of neural stem cell ablation in aged mice. **i** GFAP-TK + GCV mice had fewer BrdU⁺ cells in the SVZ, validating stem cell ablation. ****t*(15) ≥ 9.36, *p* < 0.0001. Two-tailed *t* test. *n* = 6 control, 11 GFAP-TK + GCV mice. BrdU was given twice per day for two days prior to stroke. **j** Representative images of BrdU immunostaining in the SVZ. **k** Both aged controls and GFAP-TK + GCV mice showed poor recovery following stroke. There was no difference between groups. Group main effect F(1,15) = 0.17, *p* = 0.690. Two-way ANOVA. **l** Lesion volume was not different between groups (t(15) = 0.50, *p* = 0.628, two-tailed *t* test). *n* = 6 control, 11 GFAP-TK + GCV mice. **m** Lesion reconstruction. Darker shades represent more overlap between animals. Data are presented as mean ± SEM. Where individual datapoints are shown, datapoints representing males are shown as circles; datapoints representing females are shown as squares. Source data are provided as a Source Data file.

Lineage tracing in *Nestin^CreERT2*; *Vegf^fl/fl*; Ai14 mice showed a near complete loss of VEGF in tdTomato⁺ cells in peri-infarct cortex, but no change in the number of migratory cells relative to controls at two weeks post-stroke (Fig. 6b–e). VEGF cKO did not affect the differentiation of SVZ-derived cells or affect their expression of other trophic factors (Supplementary Fig. 7). Recovery of forelimb motor function was significantly worse in VEGF cKO mice relative to controls as measured with the single-seed reaching task (Fig. 6f). Lesion size and location were not different between groups (Supplementary Fig. 8). Vascular remodeling following stroke is seen by increases in peri-infarct vascular density[3,7]. While vessel density in peri-infarct cortex was increased in control mice relative to the contralateral hemisphere, this was not seen in VEGF cKO mice, and peri-infarct vessel density in VEGF cKO mice was reduced relative to controls (Fig. 6g, h). We injected mice with BrdU daily during the peak in angiogenesis from days 5–10 after stroke to quantify new vessel formation. The number of new endothelial cells (BrdU⁺ERG⁺) was significantly reduced in VEGF cKO mice, confirming impaired angiogenesis in mice lacking VEGF in SVZ-derived cells. The finding that the number of lineage traced cells that localized to peri-infarct cortex was unaffected by VEGF cKO suggests that vascular remodeling is not required for the migration of cells from the SVZ, but is instead a consequence of their production of VEGF.

In a separate experiment, we examined the effects of conditional *Vegf* deletion on dendritic spine density after stroke (Fig. 6k). Peri-infarct layer V cortical pyramidal neurons were labeled by injections of AAV5-CaMKIIa-eGFP (Fig. 6l). Four weeks post-stroke, we quantified spine density on apical dendrites in layer II/III. We previously found that changes in spine density in this region parallel functional recovery[2]. Spine density was significantly reduced in VEGF cKO mice (Fig. 6m, n). Altogether, our results identify VEGF produced by SVZ-derived cells as a key driver of repair and recovery after stroke. More broadly, our findings position newborn cells arising from the SVZ in response to injury as a unique cellular source of trophic support that instructs neural repair.

### AAV-mediated expression of VEGF in peri-infarct cortex enhances recovery in mice with arrested cytogenesis

We next tested whether replacement of VEGF would be sufficient to enhance recovery in mice in which cytogenesis was arrested. *GFAP-TK* mice were trained on the single-seed reaching task and administered GCV to ablate neural stem cells. Immediately after stroke, mice were injected with either AAV5-EF1α-VEGF-P2A-eGFP (AAV-VEGF-eGFP), to induce VEGF and eGFP expression, or AAV5-EF1α-eGFP (AAV-eGFP), to induce only eGFP expression, into layer V of peri-infarct cortex (Fig. 7a–c). AAV-VEGF-eGFP induced rapid and sustained motor recovery as measured by the single-seed reaching task, whereas AAV-eGFP injected mice showed little improvement up to four weeks post-stroke (Fig. 7d). Lesion size and location were not different between groups (Supplementary Fig. 9). We examined vascular density in homotopic contralateral and peri-infarct cortex 28 days post-stroke.

Vascular density was significantly greater in peri-infarct cortex of AAV-VEGF-eGFP injected mice relative to AAV-eGFP mice (Fig. 7e, f). Moreover, there was no difference in vascular density between contralateral and peri-infarct cortex in the AAV-eGFP group, indicating a failure of neovascularization (t(22) = 0.97, *p* = 0.344). Mice were given daily injections of BrdU during days 5–10 post-stroke to label new blood vessels. AAV-VEGF-eGFP mice had substantially more angiogenesis in peri-infarct cortex than AAV-eGFP mice as measured by the number of BrdU⁺ERG⁺ nuclei (Fig. 7g, h). Finally, we examined spine density of eGFP-expressing pyramidal neurons on apical dendrites in layer II/III. Spine density was significantly higher in mice given AAV-VEGF-eGFP (Fig. 7i, j). In wildtype mice subjected to a sham stroke procedure, AAV-VEGF-eGFP increased vessel density but did not affect motor function (Supplementary Fig. 10). Since the ubiquitous promoter in EF1α-VEGF-eGFP induces VEGF expression across cell types, it was unclear whether the increase in spine density we observed was due to cell-autonomous or -extrinsic actions of VEGF on neurons. We generated a GFAP-VEGF AAV and found that selective expression of VEGF in astrocytes was also sufficient to increase spine density in peri-infarct cortex (Supplementary Fig. 11). Overall, our findings indicate that replacement of VEGF is sufficient to enhance repair and recovery in mice lacking cytogenesis. More broadly, these findings suggest that replacement of factors produced by SVZ-derived cells may constitute an effective therapy.

### CXCL12-CXCR4 signaling directs migration of cells arising from the SVZ to peri-infarct cortex

Having established the identity and functions of SVZ-derived cells after cortical stroke, we next sought to better understand what signals drive the ectopic migratory response. Given the close interactions between SVZ-derived cells and vasculature, we focused on signaling cues generated from endothelial cells after stroke. Gene expression data from a recent study that profiled changes in vasculature after stroke identified cytokine-receptor interactions, including CXCL12-CXCR4 signaling, as enriched pathways[61]. CXCL12 is a well-characterized chemotactic factor that governs migration of a variety of precursor cells throughout many organs. Furthermore, CXCL12 signaling via its receptor CXCR4 regulates migration of neuroblasts during development and after stroke[12,62], and migration of transplanted precursor cells after stroke[63]. Based on this profiling data, we first validated the induction of CXCL12 after stroke and observed a substantial increase in protein level expression in peri-infarct cortex compared to uninjured cortex (Fig. 8a, b). We also found that CXCL12 expression was most evident in peri-infarct vasculature rather than other cell types (Fig. 8c). In addition, CXCR4, the CXCL12 receptor, was expressed by lineage traced cells arising from the SVZ after stroke (Fig. 8d). These findings suggest that CXCL12 produced by peri-infarct blood vessels could direct migration of SVZ-derived cells via CXCR4. We next tested whether antagonizing CXCR4 affected the migration of lineage traced cells towards peri-infarct cortex. We administered tamoxifen to Nestin-Cre^ERT2; Ai14

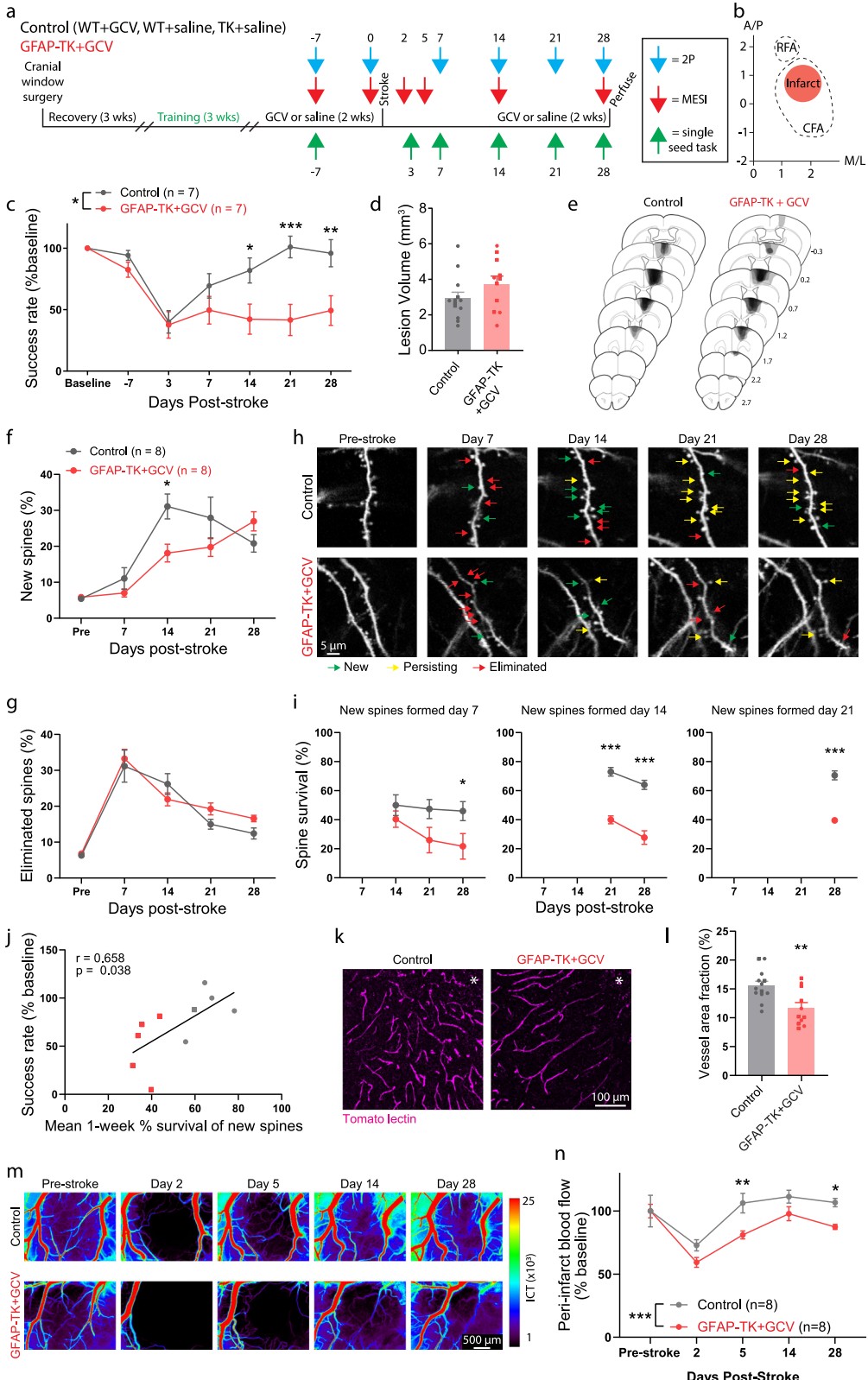

mice to label SVZ progeny. Then, we treated mice with the CXCR4 antagonist Plerixafor or saline twice daily from the day of stroke until euthanasia two weeks later (Fig. 8e). We found that CXCR4 antagonism significantly reduced the number of tdTomato+ cells that localized to peri-infarct cortex (Fig. 8f, g). These findings provide evidence that CXCL12-CXCR4 signaling contributes to the ectopic migration of SVZ-derived cells after stroke. Since the migration of

cells was not completely stopped by Plerixafor treatment, it is likely that other mechanisms also contribute to this ectopic migration. Based on our findings, we propose the following model: vasculature surrounding the infarct produces guidance cues, such as CXCL12, that directs the ectopic migration of cells from the SVZ. In turn, these SVZ-derived cells produce VEGF, and perhaps other factors, to drive repair processes in the residual tissue surrounding the infarct.

**Fig. 4 | SVZ cytogenesis supports neuronal and vascular remodeling.**
**a** Experimental design. **b** Schematic illustrating infarct placement relative to caudal (CFA) and rostral (RFA) forelimb areas in motor cortex. Axes indicate mm relative to Bregma. **c** Performance on the reaching task. GFAP-TK + GCV mice showed significantly worse recovery relative to control mice (Two-way ANOVA, significant group x time interaction, $F_{(6, 72)} = 4.7$, $p < 0.001$). *$p = 0.026$, **$p = 0.005$, ***$p = 0.0002$, post-hoc Bonferroni tests. **d** Lesion volume was not different between groups ($t_{(22)} = 1.5$, $p = 0.159$). Two-tailed $t$ test. $n = 13$ control, 11 GFAP-TK + GCV mice. **e** Lesion reconstructions. **f–j** Altered spine dynamics. **f** New spine formation was significantly higher in control mice on day 14 (Two-factor mixed-effects model, significant time x group interaction, $F_{(4, 52)} = 4.1$, $p = 0.006$). *$p = 0.011$, Sidak's multiple comparison tests between groups for each day. **g** Spine elimination was not significantly different (Two-factor mixed-effects model, group effect, $F_{(1, 66)} = 1.2$, $p = 0.285$). $n = 8$ mice/group. **h** Longitudinal images of dendritic spines illustrating formation (green), persistence of new spines (yellow), and elimination (red). **i** Survival of new spines was reduced in GFAP-TK + GCV mice. Two-factor mixed-effects model and post-hoc Sidak's multiple comparison tests, *$p = 0.038$, ***$p < 0.0001$. $n = 8$ mice/group. **j** Spine survival was correlated with behavioral performance on the final day of testing. Linear regression, $F_{(1, 8)} = 6.1$, $p = 0.038$. $n = 5$ mice/group. **k** Representative images of peri-infarct vasculature. **l** Peri-infarct vessel density was reduced in GFAP-TK + GCV mice (**$t_{(22)} = 3.3$, $p = 0.004$). Two-tailed $t$ test. $n = 13$ control, $n = 11$ GFAP-TK + GCV mice. **m** Representative blood flow maps. **n** Peri-infarct blood flow was reduced in GFAP-TK + GCV mice (Two-way repeated measures ANOVA and post-hoc Sidak's multiple comparison tests, group effect $F_{(1, 14)} = 32.98$, $p < 0.0001$). *$p = 0.020$, **$p = 0.0014$. Data are presented as mean ± SEM. Where individual datapoints are shown, datapoints representing males are shown as circles; datapoints representing females are shown as squares. Source data are provided as a Source Data file.

## Discussion

Our study revealed that a previously underappreciated class of cells, undifferentiated precursors, constitutes the majority of cells that arise from the SVZ and migrate towards the site of injury following stroke. We found that reducing SVZ cytogenesis, by neural stem cell ablation or aging, leads to poor functional recovery. Moreover, synaptic and vascular repair were disrupted in mice with deficient cytogenesis. SVZ-derived cells produced trophic factors, most notably VEGF. Loss-of-function experiments demonstrated that VEGF produced by SVZ-derived cells is crucial for effective repair and recovery. In addition, gain-of-function experiments showed that replacement of VEGF was sufficient to enhance recovery in mice lacking cytogenesis. Finally, we showed that CXCL12-CXCR4 signaling is a mechanism for the ectopic migration of SVZ-derived cells. We conclude that trophic support from SVZ-derived cells drives neural repair and functional recovery after stroke. Thus, newborn cells migrating from the SVZ in response to injury enable recovery by acting as a unique source of trophic cues that instruct neural repair.

With lineage tracing of adult neural stem cells and extensive phenotyping, we identified undifferentiated precursors as the largest subpopulation of SVZ-derived cells after stroke. Undifferentiated neural precursor cells may the better suited than differentiated neural cell types to facilitate neural repair. In culture, neural precursors secrete factors that facilitate vessel formation and neuronal outgrowth[48,57,64]. In addition, transplantation of neural stem cells enhances repair and recovery in models of stroke without differentiation of transplanted cells[43,44,46–48]. Collectively, these studies illustrate the reparative abilities of undifferentiated precursors. Importantly, these past studies have identified numerous factors produced by precursor cells depending on context. It is possible that multiple factors produced by SVZ-derived cells promote recovery after stroke. This is suggested by our finding that recovery is worse in mice with ablated neural stem cells compared with VEGF cKO mice. Thus, future studies could examine other molecular targets. It is also conceivable that SVZ-derived cells could induce the expression of growth promoting factors in resident cortical cells, but this remains to be determined.

Many past studies of post-stroke cytogenesis have examined the production of new neurons (i.e., "neurogenesis"). We found that new neurons represented the least common cell type among lineage traced SVZ-derived cells after stroke. Therefore, our study provides evidence that the production of new neurons after stroke is a relatively small component of the cytogenic response, suggesting that other cell types —such as undifferentiated precursors or astrocytes—might be more important for recovery. The finding that cytogenesis promotes synaptic and vascular plasticity via trophic support in order to facilitate recovery demonstrates a function of post-stroke cytogenesis that may be independent of the production of new neurons. We did not find evidence of delayed differentiation of lineage traced cells out to 8 weeks post-stroke, which is well after the period of substantial functional improvement. It remains possible that there is some delayed differentiation of these cells at even later time points.

While neuroblast migration in the adult brain is well documented, migration of neural stem cells and undifferentiated precursors is rare[12,20]. However, it is unlikely that migratory neuroblasts are the source of SVZ-derived cells that migrate to peri-infarct cortex. For instance, we found few neuroblasts along the migration route from the SVZ to the infarct across multiple time points after stroke. The de novo migration of precursor cells appears to be due to the unique and high expression of CXCL12 in peri-infarct regions.

We identified VEGF produced by SVZ-derived cells as critical for effective repair and recovery after stroke. The potent angiogenic effects of VEGF have been well studied[58,59], but its effects on neuronal growth are less well understood. It is possible that VEGF produced by SVZ-derived cells directly enhanced neuronal outgrowth and synapse formation since ex vivo evidence suggests that VEGF increases neuronal complexity[57]. An additional possibility is that restored blood flow due to VEGF-mediated angiogenesis provided metabolic support for neuronal growth[7,65]. Indeed, growth of new blood vessels after stroke supports local blood flow increases[3], which in turn contributes to the degree of local synaptogenesis[51]. It has been reported that VEGF can have neuroprotective effects and that depleting neural stem cells can exacerbate lesion size after stroke[15,60]. However, we did not detect any differences in lesion size between groups across our experiments using the photothrombosis model, which has a relatively small penumbra compared to other models[66]. It is possible that these effects are only seen in models with large ischemic penumbras where more tissue is at risk of delayed death. It is also possible that the nature of the SVZ migratory response and the factors produced by these cells might be different in other stroke models.

Further work is needed to understand how the fate of individual cells is decided among the total population of newborn cells arising from the SVZ. Principles of neural development may apply to cell fate decisions during post-injury cytogenesis in adulthood. For example, the transcription factor NFIA controls gliogenesis during development[67], and is also necessary for SVZ astrogenesis after stroke[68]. In addition, NFIA inhibits neurogenesis via Notch effectors[67,69]. Accordingly, interfering with Notch signaling in neural stem cells biases SVZ progeny towards a neuronal fate after stroke[19]. Additional work will be needed to clarify the mechanisms that dictate the identity of SVZ cells responding to stroke, and whether different cell types have distinct reparative functions.

Our finding that aging reduces SVZ cytogenesis could be translationally relevant given that stroke incidence increases with age[35]. We observed the loss of stroke-induced SVZ proliferation and precursor cell pool expansion in aged mice, which suggests deficient activation of neural precursor cells with aging. Inflammatory signals have been implicated in controlling quiescence/activation of precursors. Chronic inflammatory signals, including interferons, promote stem cell quiescence in aging[39]. By contrast, acute interferon signaling after stroke

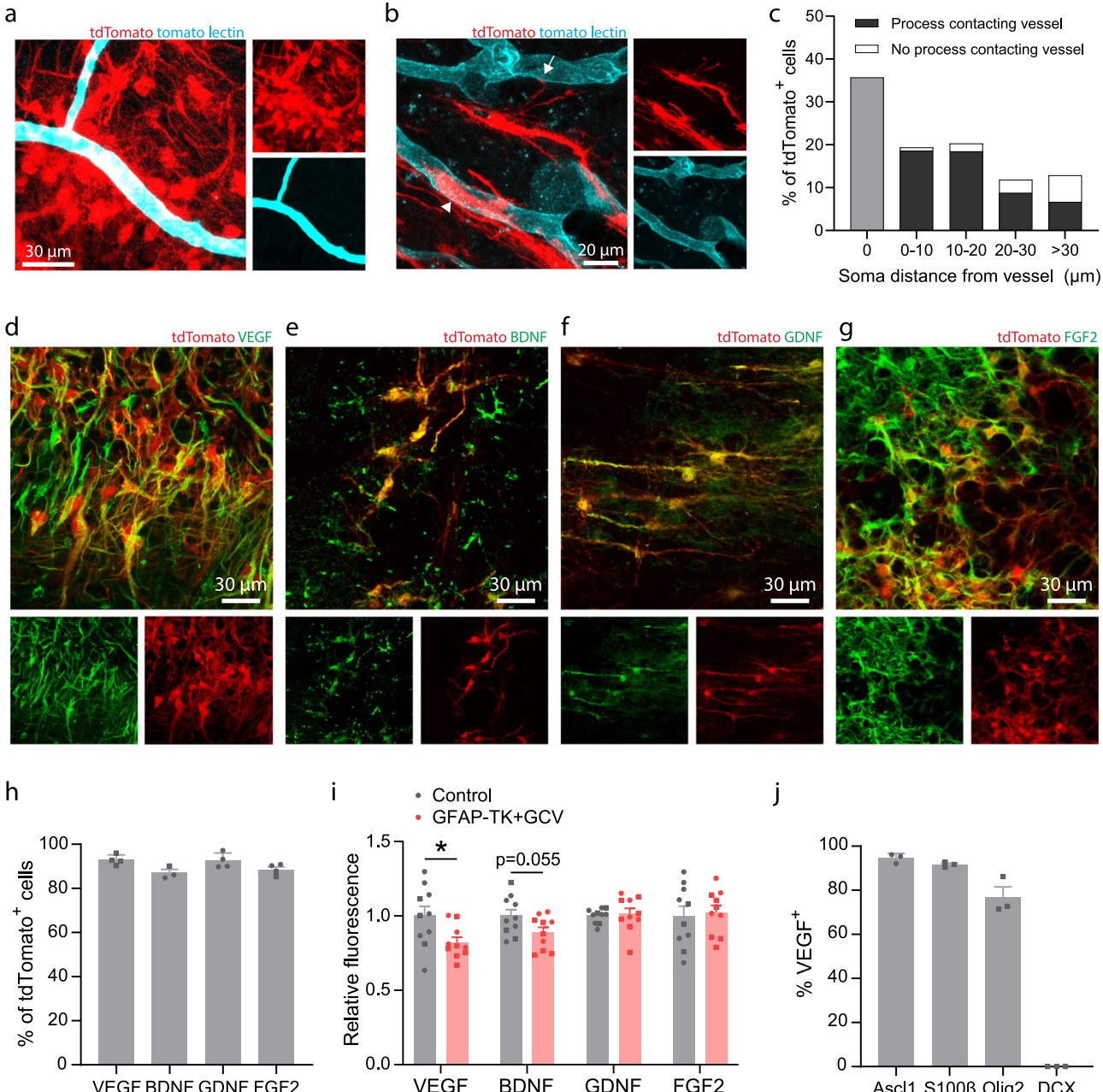

**Fig. 5 | SVZ-derived cells interact with vasculature and produce trophic factors.**
**a** Confocal image showing clustering of lineage traced cells around blood vessels in peri-infarct cortex. **b** tdTomato-expressing cells were frequently observed with cell bodies abutting vessels (arrowhead) or extending processes that terminated on nearby vessels (arrow). Images in (**a**, **b**) are representative of experiments repeated in 3 mice. **c** Quantification of SVZ-derived cell interaction with vasculature in peri-infarct cortex two weeks post-stroke. Data are from 661 cells across 3 *Nestin*CreERT2; Ai14 mice. **d–g** Confocal images from peri-infarct cortex showing expression of the trophic factors VEGF (**d**), BDNF (**e**), GDNF (**f**), and FGF2 (**g**) in tdTomato+ cells. **h** Quantification of trophic factor expression in tdTomato+ cells. *n* = 4 mice for VEGF, GDNF, and FGF2; *n* = 3 mice for BDNF. **i** Quantification of

trophic factor expression in peri-infarct cortex four weeks post-stroke between control mice and GFAP-TK + GCV mice (*n* = 10/group). Fluorescence intensity is reported relative to controls. VEGF protein fluorescence was significantly reduced in GFAP-TK + GCV mice. *t(18) = 2.4, *p* = 0.025. Two-tailed *t* test. **j** Quantification of VEGF expression by phenotype in lineage traced cells. *n* = 3 mice per marker. VEGF was expressed by Ascl1+ precursors, S100β+ astrocytes, and Olig2+ oligodendrocyte-lineage cells, but not by neuronal lineage cells (DCX+). Data are presented as mean ± SEM. Where individual datapoints are shown, datapoints representing males are shown as circles; datapoints representing females are shown as squares. Source data are provided as a Source Data file.

stimulates precursor activation[28,70]. Therefore, inflammatory signaling pathways may be a target to restore cytogenesis in aged animals. Notably, there is evidence that precursor cells can attenuate inflammation and have protective effects after injury, suggesting reciprocal interactions between inflammation and precursor cells[44,55]. We also observed a reduction in the number of cells localized in peri-infarct regions in aged mice that was disproportionate relative to the

diminishment of SVZ cytogenesis. Thus, there may also be an age-dependent reduction in either the migratory ability of SVZ cells or the expression of migratory cues at the site of injury.

The persistence of neural stem cells and cytogenesis in adult humans is debated. While it is generally accepted to occur perinatally and in children, there is evidence for and against SVZ cytogenesis in the healthy and injured adult human brain[71–77]. If cytogenesis declines

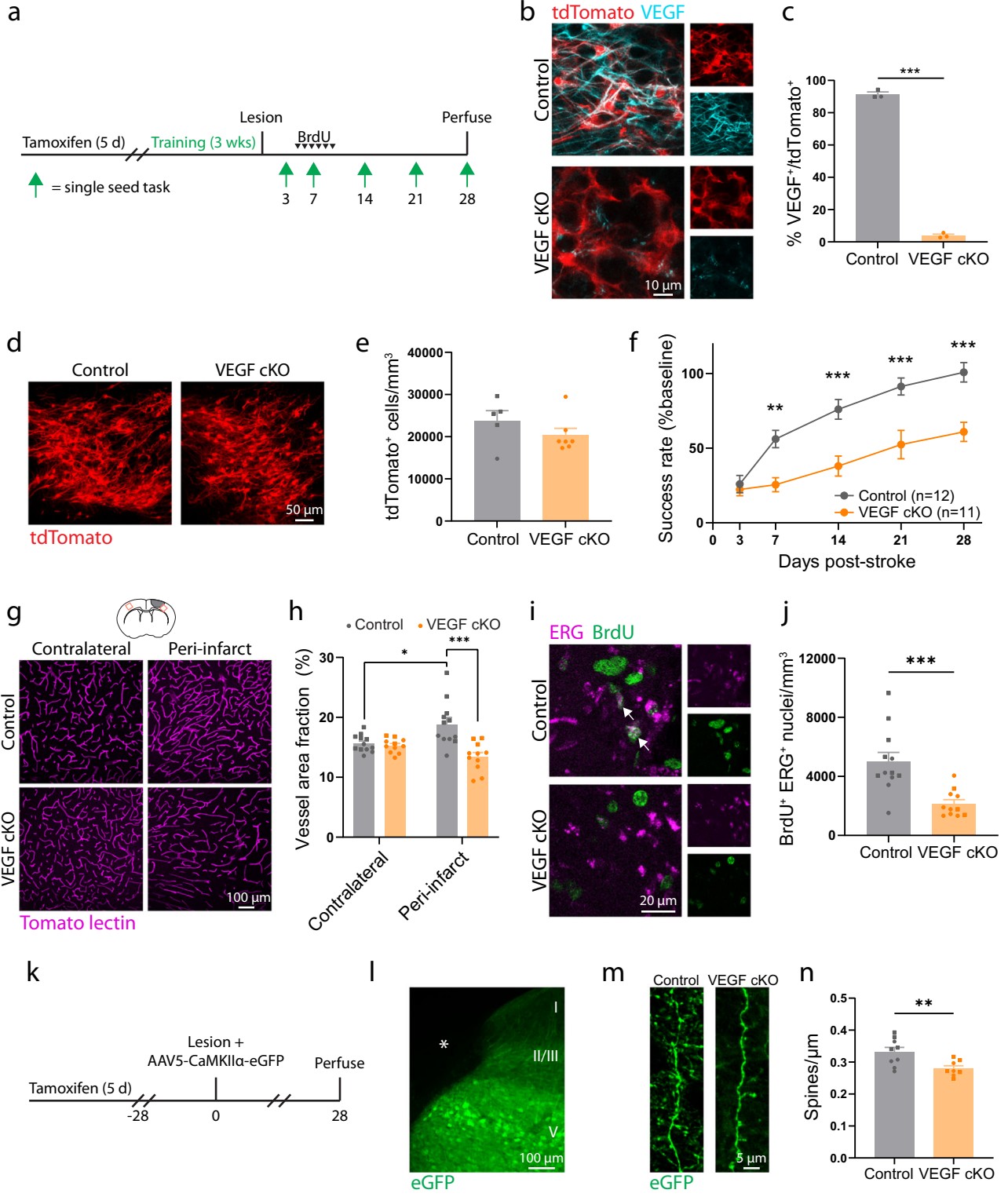

in aging humans, as in rodents, our study provides rationale for several treatment strategies: slowing neural stem cell decline, enhancing neural stem cell activation, and replacing factors produced by the neural stem cell lineage.

We have demonstrated that the SVZ cytogenic response to stroke primarily produces undifferentiated precursors that localize to peri-infarct regions – the site of neural repair. This migration is driven at least in part via CXCL12-CXCR4 signaling. SVZ-derived cells produce VEGF that is critical for effective vascular and synaptic plasticity, and ultimately behavioral recovery. Our findings support a model whereby

stroke induces the expression of chemoattractive cues to direct ectopic migration of cells from the SVZ towards the site of injury, and, in turn, SVZ-derived cells promote recovery via trophic support. These findings provide insight into a fundamental brain repair process and may be relevant for informing treatment strategies.

## Methods

### Subjects and experimental design

Animal use was approved (AUP-2018-00283) by the Institutional Animal Care and Use Committees at the University of Texas at Austin or

**Fig. 6 | Adult neural stem cell lineage-specific deletion of *Vegf* impedes recovery and repair. a** Timeline of experiments in control and VEGF mice. **b** Confocal images demonstrating VEGF loss in tdTomato⁺ cells in *Nestin*^CreERT2+/-; *VEGF*^fl/fl; Ai14 mice. **c** Efficiency of VEGF cKO. $n = 3$ mice/group. ***t(4) = 51.23, $p < 0.0001$. Two-tailed $t$ test. **d** Representative images of tdTomato⁺ cells in peri-infarct cortex two weeks post-stroke. **e** There was no difference in peri-infarct tdTomato⁺ cell density between controls and VEGF cKO mice, indicating that the migratory response was unaffected. $n = 5$ controls, $n = 7$ VEGF cKO. t(10) = 1.18, $p = 0.266$. Two-tailed $t$ test. **f** VEGF cKO significantly impaired motor recovery measured with the single-seed reaching task ($n = 12$ control, $n = 11$ VEGF cKO). Two-way repeated measures ANOVA and post-hoc Sidak's tests between groups for each timepoint. Time x group interaction F(4, 84) = 9.2, $p < 0.001$. **$p = 0.005$, ***$p \leq 0.0002$. **g** Representative confocal images and quantification (**h**) of vasculature. *t(22) = 2.8, $p = 0.011$. ***t(21) = 4.1, $p = 0.0005$. Two-tailed $t$ tests. $n = 12$ controls, $n = 11$ VEGF cKO mice.

**i** Confocal images of new endothelial cells in peri-infarct cortex. **j** The number of BrdU⁺ ERG⁺ nuclei was significantly less in VEGF cKO mice. ***t(15.0) = 4.2, $p = 0.0008$, Welch's corrected two-way $t$ test. $n = 12$ controls, $n = 11$ VEGF cKO mice. **k** Experimental timeline. **l** Image shows AAV5-CaMKiiα-eGFP labeling in peri-infarct cortex (representative of 17 mice). Asterisk indicates lesion core. **m** Representative images and quantification (**n**) of dendritic spine density. Spine density was significantly lower in VEGF cKO mice. **t(15) = 3.0, $p = 0.009$. Two-tailed t test. Apical dendrites were sampled from layer II/III between 100 and 700 μm from the infarct border. 2631 spines were counted along 7.9 mm total length of dendrite in controls ($n = 9$ mice). 1934 spines were counted along 6.9 mm total length of dendrite in cKO mice ($n = 8$ mice). Data are presented as mean ± SEM. Individual datapoints representing males are shown as circles; datapoints representing females are shown as squares. Source data are provided as a Source Data file.

Baylor College of Medicine. Young adult (3–6 months) and aged (12–16 months) mice of both sexes were used. All mice were on a predominantly C57BL/6 background. Transgenic strains were Ai14 (*Rosa26*^CAG-LSL-tdTomato, JAX #007914), *Nestin*^CreERT2 (JAX #016261), *Ascl1*^CreERT2 (JAX #012882), *Rosa26*^CAG-LSL-Sun1-sfGFP (JAX #021039), *GFAP-TK* (JAX #005698), *Thy1-GFP* M-line (JAX #007788), floxed *Vegfa* (Genentech)[58] (Supplementary Data 1). Mice were bred locally. Animals were housed 2–5 per cage with free access to food and water, except during periods of restricted feeding for behavioral training and assessment. Animals were randomized to groups except when group assignment was dependent on genotype. Experimentation and analysis were done blinded to group allocation. Experiments consisted of 1–5 cohorts of animals. Sample sizes were based on past work using similar methods[1–3,7,19].

### Drug administration
100 mg/kg of 20 mg/mL tamoxifen dissolved in corn oil was given (i.p.) daily for 5 consecutive days. 100 mg/kg of 10 mg/mL BrdU dissolved in saline was given (i.p.) once or twice per day for 2 or 6 consecutive days. Ganciclovir dissolved in saline (typically 25–30 mg/mL, adjusted for pump rate) was delivered continuously for 14 days via subcutaneous osmotic pumps (Azlet) at a rate of 6.25 μg/hr. In some experiments, a second course of ganciclovir was given beginning two weeks after stroke to maintain stem cell ablation. We did not observe any differences in animal behavior between the single and double dose regimens. We found no differences in functional recovery between the following control groups: wildtype+saline, wildtype+GCV, GFAP-TK +saline (Fig. 2l). These conditions were combined into a single control group where exact conditions are not otherwise specified. Plerixafor was dissolved in saline at a concentration of 0.2 mg/mL and was injected s.c. at 1.25 mg/kg twice daily, as previously described[78], starting on the day of stroke until euthanasia 14 days later.

### Cranial window implantation
Chronic glass cranial windows were placed over forelimb motor cortex[2,3,7]. Isoflurane (3% induction, 1–2% maintenance) in oxygen was used for anesthesia. Body temperature was maintained with a heated pad for the duration of anesthesia. Circular craniotomies (~4.5 mm diameter) were made 1.5 mm lateral from Bregma with a dental drill. The craniotomies were regularly irrigated with sterile saline. 4 mm glass windows (Warner Instruments) were secured in place with cyanoacrylate, and exposed skull was covered with dental cement. Animals were allowed at least 2 weeks to recover before the start of imaging. Carprofen (5 mg/kg, i.p.) was given daily for 7 days to minimize inflammation.

### Ischemic stroke
To model stroke, unilateral photothrombotic lesions were induced in the forelimb region of motor cortex[7,32]. Isoflurane (3% induction, 1–2% maintenance) in oxygen was used for anesthesia. Body temperature was maintained with a heated pad for the duration of anesthesia. Stroke was induced through the intact skull by making a scalp incision, administering rose bengal (0.15 mL, 15 mg/mL, i.p.), and illuminating the skull 2 mm lateral from Bregma with a surgical lamp (Schott KL 200) for 15 minutes though a 3 mm aperture. For animals with cranial windows, penetrating arterioles supplying motor cortex (approximately 1.5 mm lateral and 0.5 mm anterior from Bregma) were identified by live speckle contrast imaging (see Blood flow imaging section below), and subsequently targeted with a 20 mW 532 nm laser (spot size approximately 300 μm) for 15 minutes after administering rose Bengal (0.2 mL, 15 mg/mL, i.p.)[3]. Sham stroke procedures involved omitting either illumination or rose bengal. For experiments involving behavioral testing, stroke was induced in the hemisphere contralateral to the preferred paw.

### Virus injections
Isoflurane (3% induction, 1–2% maintenance) in oxygen was used for anesthesia. Body temperature was maintained with a heated pad for the duration of anesthesia. Cortical layer V was targeted for virus injections with a Drummond Nanoject II microinjector through a pulled pipette. Injections were made 0.7 mm below the pial surface at three locations relative to Bregma: 2 mm anterior, 2 mm lateral; 0.5 mm anterior, 3.2 mm lateral; and 1 mm posterior, 2.8 mm lateral. 230 nL was injected per site at a rate of 46 nL/min. The pipette was left in place for 2 minutes after the final injection at each site before it was slowly removed. AAV2/ 9-GFAP-VEGF was constructed by Gibson assembly using TRE-VEGFA EF1a-rtTA3 (Addgene #195881) and pAAV.GfaABC1D.PI.Lck-GFP.SV40 (Addgene #105598) plasmids and was packaged by the Optogenetics and Viral Vectors Core at the Jan and Dan Duncan Neurological Research Institute. See Supplementary Data 1 for details on viruses.

### 2-photon imaging of dendritic spines
Mice were anesthetized with isoflurane (3% induction, -1.5% maintenance) in oxygen and head-fixed to minimize breathing artifacts. Body temperature was maintained with a heated pad for the duration of anesthesia. Imaging was done with a Prairie Ultima 2-photon microscope with a Ti:Sapphire laser (MaiTai, Spectra Physics) tuned to 870 nm. Image stacks were acquired with 512 ×512 pixel resolution and 0.7 μm z step size using a water-immersion 20×/1.0 (Olympus) objective. 4x magnification yielded a 117.2 μm × 117.2 μm field of view. Images were collected using Prairie View (5.3) software.

Imaging was done weekly, including two pre-stroke and four post-stroke imaging sessions. During pre-stroke imaging, typically 6-10 regions of unobstructed dendrites were imaged to a depth of -150 μm. Regions were selected based on a predicted proximity of <1 mm from the infarct. After stroke, the infarct border was identified from blood flow maps and loss of GFP fluorescence (Supplementary Fig. 5). Regions within 700 μm of the infarct border were re-imaged at subsequent time points in order to track spine dynamics during recovery. In some animals, additional imaging regions were added after stroke. Time lapse images of dendrites >30 μm in length with clearly visible spines were

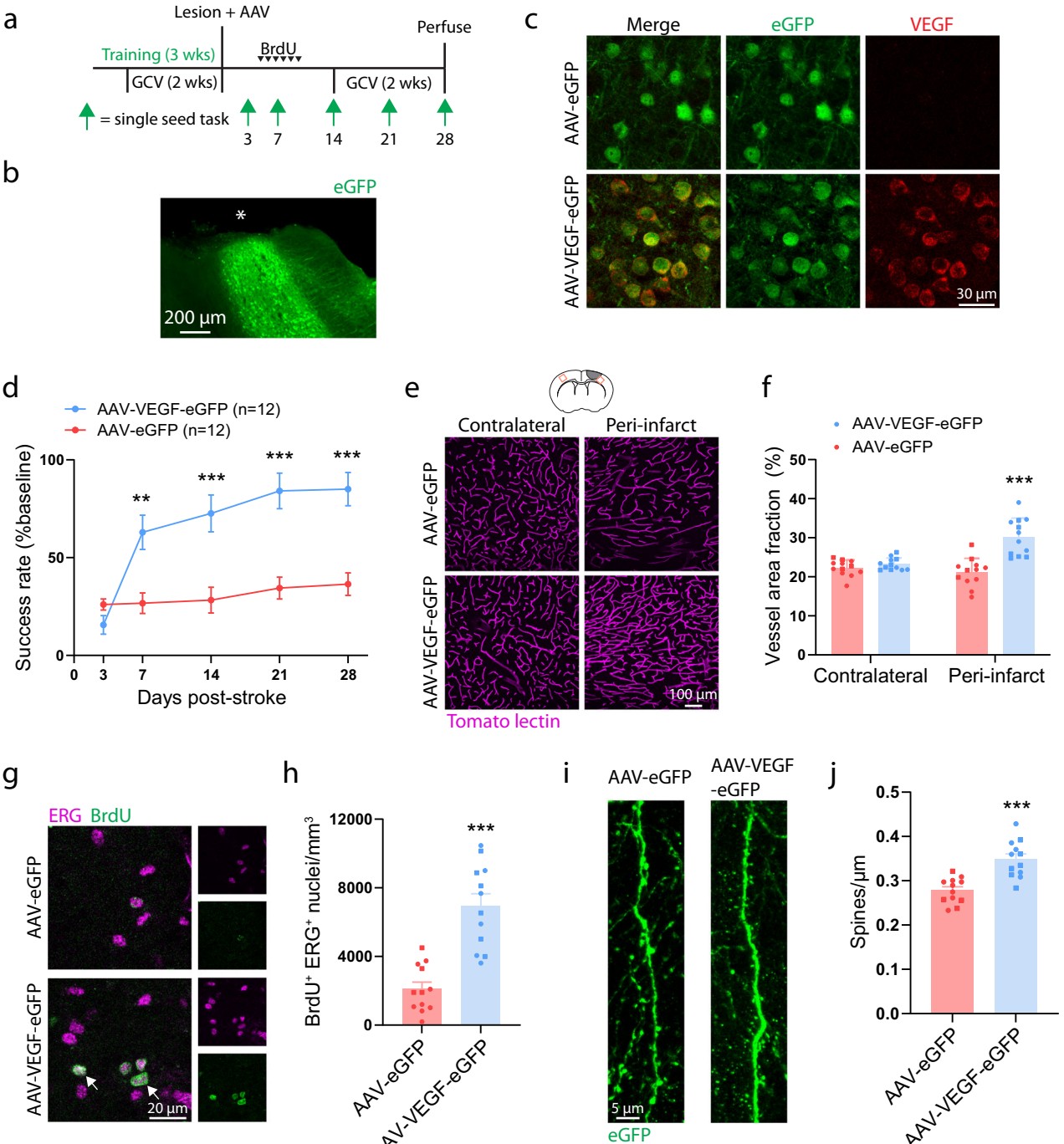

**Fig. 7 | VEGF rescues poor recovery caused by neural stem cell ablation.**
**a** Experimental timeline. Both groups used GFAP-TK mice given GCV to ablate neural stem cells ($n = 12$ per group). Animals were injected with either AAV5-Ef1α-eGFP (AAV-eGFP) or AAV5-Ef1α-VEGF-P2A-eGFP (AAV-VEGF-eGFP) in layer V of peri-infarct cortex. **b** Image illustrating AAV targeting of peri-infarct cortex (representative of 24 mice). Asterisk indicates lesion core. **c** Confocal images validating that AAV-VEGF-eGFP induces VEGF expression (representative of 3 mice per condition). **d** AAV-VEGF-eGFP improved motor recovery on the single seed reaching task. Significant time x group interaction $F_{(4, 88)} = 15.9$, $p < 0.0001$. $**p = 0.0018$, $***p < 0.0001$, two-way repeated measures ANOVA and post-hoc Sidak's multiple comparison tests. **e** Representative confocal images and quantification (**F**) of vasculature show that AAV-VEGF-eGFP increased peri-infarct vessel density relative to AAV-eGFP mice. $***t_{(22)} = 5.3$, $p < 0.0001$, two-tailed $t$ test. $n = 12$

mice/group. **g** Representative confocal images of new endothelial cells (BrdU⁺ ERG⁺, arrows) in peri-infarct cortex. **h** The number of BrdU⁺ ERG⁺ nuclei was significantly greater in AAV-VEGF-eGFP mice. $***t_{(22)} = 6.0$, $p < 0.0001$. Two-tailed $t$ test. $n = 12$ mice/group. **i** Representative images and quantification (**j**) of dendritic spine density. Spine density was significantly higher in AAV-VEGF-eGFP mice. $***t_{(22)} = 4.8$, $p < 0.0001$. Two-tailed $t$ test. $n = 12$ mice/group. Apical dendrites were sampled from layer II/III between 100 and 800 μm from the infarct border. 2825 spines were counted along 10.2 mm total length of dendrite in AAV-eGFP mice. 3231 spines were counted along 9.3 mm total length of dendrite in AAV-VEGF-eGFP mice. Data are presented as mean ± SEM. Where individual datapoints are shown, datapoints representing males are shown as circles; datapoints representing females are shown as squares. Source data are provided as a Source Data file.

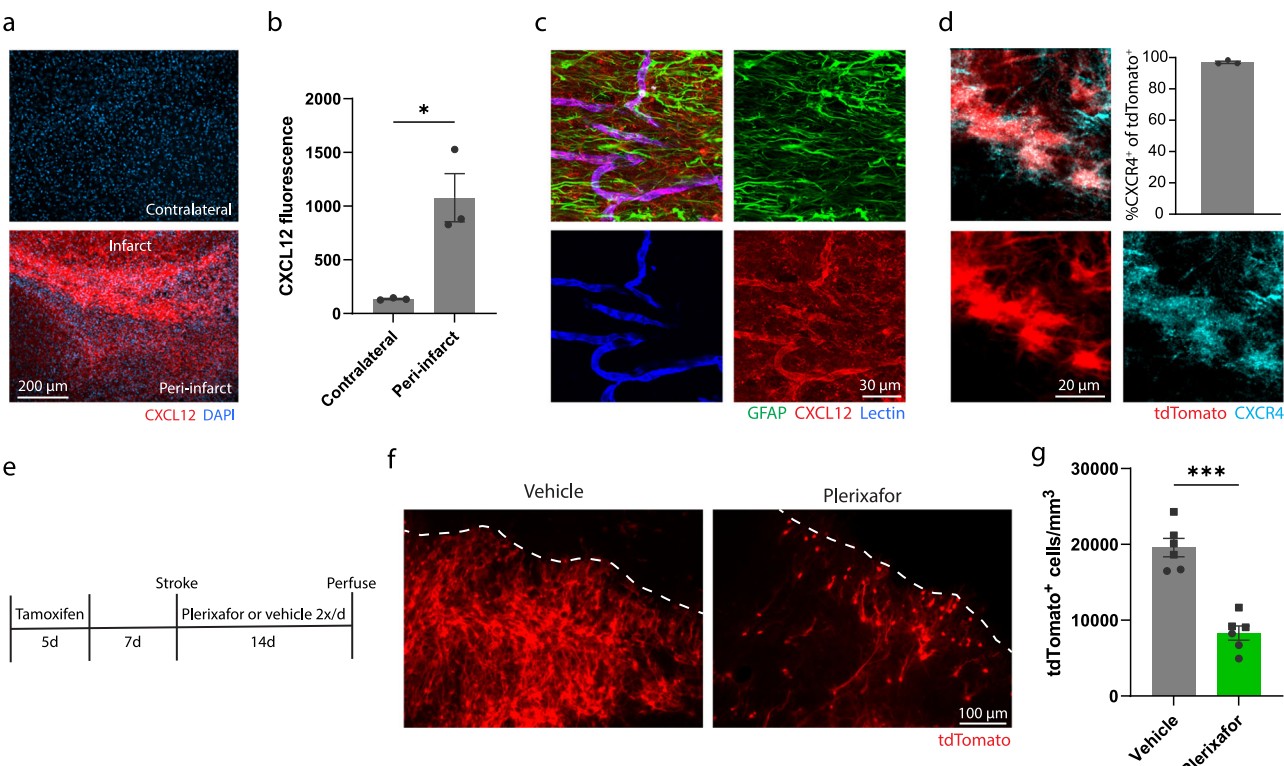

**Fig. 8 | CXCL12-CXCR4 signaling contributes to the migration of cells from the SVZ to peri-infarct cortex. a** Fluorescence images showing upregulation of the chemokine CXCL12 in peri-infarct cortex. **b** Quantification of CXCL12 fluorescence intensity in contralateral and peri-infarct cortex 7 days post-stroke. *t(4) = 4.2. p = 0.014, two-tailed t test. N = 3 mice. **c** Immunostaining revealed strong expression of CXCL12 in peri-infarct vasculature (labeled with tomato lectin), but not astrocytes (labeled with GFAP). Images are representative of experiments conducted in n = 3 mice. **d** Confocal images and quantification showing that lineage traced cells migrating from the SVZ to peri-infarct cortex express the CXCL12

receptor CXCR4. n = 3 mice. **e** Experimental timeline for antagonizing CXCR4. **f** Images of lineage traced cells migrating from the SVZ to peri-infarct cortex 14 days post-stroke in mice treated with Plerixafor or vehicle. **g** Treatment with the CXCR4 antagonist Plerixafor significantly reduced the number of SVZ-derived cells that migrated to peri-infarct cortex. *t(10) = 7.4. p < 0.0001, two-tailed t test. N = 6 *Nestin^CreERT2*; Ai14 mice per group. Data are presented as mean ± SEM. Where individual datapoints are shown, datapoints representing males are shown as circles; datapoints representing females are shown as squares. Source data are provided as a Source Data file.

analyzed to quantify spine formation and elimination, and persistence of newly formed spines[2,4,6]. Z-stack images from all time points for a given region were opened in ImageJ (1.51j8). Individual spines were manually annotated and the proportion of persistent, new, and eliminated spines was calculated relative to the previous time point.

### Blood flow imaging
Blood flow was imaged through cranial windows with multi-exposure speckle imaging, a label-free, quantitative, optical method[2,3,7]. Anesthetic level was consistent across all imaging sessions (1.25% isoflurane in oxygen). Body temperature was maintained with a heated pad for the duration of anesthesia. Mice were head-fixed and the cranial window was illuminated by red light controlled by an acousto-optic modulator. Backscattered light was collected by a CCD camera. Speckle contrast values, which are defined as the ratio of the standard deviation to the mean of the intensity, were calculated. MATLAB (R2018a) was used to calculate the inverse correlation time, which is proportional to blood flow. Processed images were exported to ImageJ where regions of interest were defined with the polygon selection tool. Blood flow was measured in parenchymal regions and tracked over time[7]. Two pre-stroke images were collected to establish baseline blood flow. Post-stroke images were collected on days 2, 5, 14, and 28. Each imaging session lasted <10 mins.

### Behavioral testing
Skilled forelimb use was assessed with the single seed reaching task, which is highly sensitive to deficits caused by motor cortical damage

and is translationally relevant[2,3,33,34]. Animals were food restricted to ~90% free feeding weight to encourage reaching. The training chamber was 20 cm tall, 15 cm deep, and 8.5 cm wide. Animals could reach for millet seeds through a 4 mm wide vertical opening in the middle of the chamber wall during shaping or on the left or right side of the chamber during training and testing. First, animals were shaped on the task and the preferred paw for reaching was determined over 2–5 days. This involved allowing the animals to reach with either forelimb for pellets. The limb most frequently used for reach attempts was considered the preferred limb. Several seeds were placed in a large well during the shaping phase. One seed at a time was placed on a 1.2 cm tall platform during the training and testing phases. The seed was placed 0.5 mm perpendicular to the center of and 0.7 mm away from the vertical opening. Training was done over 15 sessions, once per day, five days per week. Each session consisted of 30 trials. For each trial, animals were allowed up to two reach attempts. A successful reach was defined as the animal grasping the seed and bringing it to its mouth. Failure was defined as missing the seed, knocking it out of the well, or releasing it before it was brought to the animal's mouth. Baseline performance was defined as the mean success rate per trial over the last three training sessions. Inclusion criteria was a minimum baseline success rate of 30%. 1 mouse (GFAP-TK + GCV) from the experiment in Figs. 2, 8 mice (n = 7 control, n = 1 GFAP-TK + GCV) from the experiment in Figs. 3, and 2 mice (n = 1 AAV-eGFP, n = 1 AAV-VEGF-eGFP) from the experiment in Supplementary Fig. 10 failed to meet this threshold and were excluded. Test sessions were done on days 3, 7, 14, 21, and 28 post-stroke.

## Histology and image analysis

Mice were euthanized by overdose with a pentobarbitol/phenytoin solution followed by perfusion with 0.1 M phosphate buffer and 4% paraformaldehyde in phosphate buffer. Brains were postfixed overnight at 4 °C in 4% paraformaldehyde in phosphate buffer. 35 μm coronal sections were collected with a vibratome (VT1000S, Leica). To label vasculature, mice were retro-orbitally injected with 0.1 mL of Dylight 594- or 649-conjuagated tomato lectin 5 minutes prior to perfusion[3,7]. To examine vascular permeability, mice were retro-orbitally injected with 0.1 mL of 50 mg/mL FITC-conjugated albumin 2 hours prior to perfusion.

To create lesion reconstructions and quantify lesion volume, one set of every fifth section was Nissl stained with toluidine blue. Lesions were reconstructed by manually tracing the lesion territory onto coronal section schematics[8]. Tracings were scanned and tracings from each group were digitally overlayed. For determining lesion volume, entire Nissl stained sections (every tenth sections) were imaged with a brightfield microscope. Images were opened in ImageJ and the polygon selection tool was used to trace the area of contralateral and residual ipsilateral cortex. Lesion volume was calculated using Cavalieri's method as the difference in volume between uninjured and injured cortex.

Immunohistochemical staining was done by washing tissue in phosphate buffered saline (PBS), blocking with 10% donkey serum in PBS with 0.25% Triton for 60 minutes, incubating with primary antibodies overnight, washing in PBS, incubating with species appropriate 405-, 488-, 594-, or 647-conjugated secondary antibodies, and washing a final time in PBS. Tissue was pretreated with 2 N HCl (30 mins) followed by 0.1 M boric acid (10 mins) when staining for BrdU. Primary antibodies and dilutions were as follows: Rabbit polyclonal anti-ASCL1 (1:1000) Abcam ab74065, Rabbit polyclonal anti-ASCL1 (1:500) Cosmo Bio CAC-SK-T01-003, Rabbit polyclonal anti-CD133 (1:1000) Abcam ab19898, Rabbit monoclonal anti-BDNF (clone EPR1292) (1:1000) Abcam ab108319, Rabbit polyclonal anti-BrdU (1:500) Abcam ab152095, Rat monoclonal anti-BrdU [clone BU1/75 (ICR1)] (1:500) Abcam ab6326, Rabbit polyclonal anti-cleaved caspase 3 (1:500) Cell Signaling #9661, Mouse monoclonal anti-CXCL12 (Clone 79018) (1:200) R&D Systems MAB350, Rabbit monoclonal anti-CXCR4 (clone UMB2) (1:250) Abcam Ab124824, Goat polyclonal anti-DCX (1:500) Santa Cruz Biotech. Sc-8066, Rabbit monoclonal anti-ERG (clone EPR3864) (1:500) Abcam ab92513, Rabbit polyclonal anti-FGF2 (1:500) Sigma F-3393, Rabbit polyclonal anti-GDNF (1:100) Abcam ab18956, Rabbit polyclonal anti-GFAP (1:1000) Dako Z0334, Chicken polyclonal anti-GFAP (1:2000) Abcam ab4674, Chicken polyclonal anti-GFP (1:5000) GeneTex GTX13970, Goat polyclonal anti-HSV thymidine kinase (1:1000) Santa Cruz Biotech. Sc-28038, Rabbit monoclonal anti-Id2 (Clone 9-2-8) (1:1000) CalBioreagents M213, Rabbit polyclonal anti-Ki67 (1:500) Abcam ab66155, Mouse monoclonal anti-Nestin (clone Rat401 (RUO)) (1:100) BD Pharmingen 556309, Mouse monoclonal anti-NeuN (clone A60) (1:500) Millipore MAB377, Rabbit monoclonal anti-NeuN (clone 27-4) (1:2000) Millipore MABN140, Rabbit polyclonal anti-Olig2 (1:1000) Millipore AB9610, Rabbit monoclonal anti-S100β (clone EP1576Y) (1:1000) Abcam ab52642, Rabbit polyclonal anti-Sox2 (1:1000) Millipore AB5603, Rabbit polyclonal anti-VEGF (1:1000) Millipore ABS82, Rabbit polyclonal anti-VEGF AF647 conjugate (1:1000) Millipore ABS82-AF647, Rabbit polyclonal anti-VEGF (1:250) Sigma 07-1420. Secondary antibodies and dilutions were as follows: Alexa Fluor 488-conjugated donkey anti-chicken (1:500) Jackson ImmunoResearch 703-545-155, Alexa Fluor 488-conjugated donkey anti-goat (1:500) Jackson ImmunoResearch 705-545-147, Alexa Fluor 594-conjugated donkey anti-goat (1:500) Jackson ImmunoResearch 705-585-147, Alexa Fluor 488-conjugated donkey anti-mouse (1:500) Jackson ImmunoResearch 715-545-151, Alexa Fluor 647-conjugated donkey anti-mouse (1:500) Jackson ImmunoResearch 715-605-151, Alexa Fluor 488-conjugated donkey anti-rabbit (1:500) Jackson ImmunoResearch 711-

545-152, Alexa Fluor 594-conjugated donkey anti-rabbit (1:500) Jackson ImmunoResearch 711-585-152, Alexa Fluor 647-conjugated donkey anti-rabbit (1:500) Jackson ImmunoResearch 711-605-152, Alexa Fluor 488-conjugated donkey anti-rat (1:500) Jackson ImmunoResearch 712-545-153, Alexa Fluor 594-conjugated donkey anti-rat (1:500) Jackson ImmunoResearch 712-585-153.

Confocal images with 1–2 μm Z-step size were collected with a Leica TCS SP5 microscope with Leica LAS software (2.7.3.9723). 20×/0.7 NA and 40×/1.0NA objectives were used. Acquisition settings were consistent between samples. Typically, 3 sections were imaged per region of interest per mouse.

ImageJ (1.51j8) was used for image analysis. Area fraction and fluorescence intensity were quantified as before[7]. For area fraction, images were binarized and the Analyze Particles tools was used to determine area fraction. For fluorescence intensity, regions of interest were outlined with a selection tool and the Measure function was used to determine intensity. The optical disector method was used to count cell density. Cell density was calculated by number of cells / (frame area × section thickness). For immunohistochemical analysis of lineage traced cell identity, images were taken and cells were examined throughout peri-infarct cortex.

## Statistical analysis

Data are expressed as mean ± S.E.M. Measurements from individual animals are shown on plots as datapoints where possible. Data were analyzed with GraphPad Prism version 9.3. Independent samples were compared with two-tailed t tests. Variance was assessed with F tests, and Welch's corrected $t$ tests were used in cases where variance was significantly different. One- and two-way ANOVAs, mixed-effects analyses, and linear regressions were used as noted in the legends. Post-hoc tests were used following significant ANOVA as noted in the legends. Details on the statistical tests used for each experiment are located in the Results and figure legends. Alpha was set at $P < 0.05$.

## Reporting summary

Further information on research design is available in the Nature Portfolio Reporting Summary linked to this article.

## Data availability

Source data are provided with this paper. Additional data supporting the findings of this study are provided in the Supplementary Information. Source data are provided with this paper.

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

## Acknowledgements

This work was performed with the support of the Mouse Genetic Engineering Facility (RRID:SCR 021927), a core facility within the Center for Biomedical Research Support at the University of Texas at Austin. AAV packaging was done by the Optogenetics and Viral Vectors Core at the Jan and Dan Duncan Neurological Research Institute. This work was supported by the following: Canadian Institutes of Health Research award DFS-157838 (MRW), National Institutes of Health grant NS108484 (AKD), National Institutes of Health grant EB011556 (AKD), National Institutes of Health grant MH102595 (MRD), National Institutes of Health grant MH117426 (MRD), National Institutes of Health grant NS056839 (TAJ).

## Author contributions

Conceptualization: M.R.W., Methodology: M.R.W., T.A.J., M.R.D., Investigation: M.R.W., S.P.L., R.L.F., N.A.D., J.L.R., M.S.N.-C., A.C., Visualization: M.R.W., S.P.L., R.L.F., Supervision: M.R.D., T.A.J., A.K.D., B.D., Writing—original draft: M.R.W., Writing—review & editing: All authors.

## Competing interests

The authors declare no competing interests.
