## [Peer Review File · Nature Communications]

Subventricular zone cytotogenesis provides trophic support for neural repair in a mouse model of strokeREVIEWER COMMENTS

Reviewer #1 (Remarks to the Author):

Williamson et al. present a study that focuses on the role of subventricular zone (SVZ)-derived cells in promoting recovery after stroke. They propose that cortical stroke triggers substantial migration of progenitors, and that these remain in an undifferentiated state long after injury has been inflicted. Using SVZ-cell ablation studies, they see a beneficial effect of these cells on tissue repair (in the form of synaptogenesis and angiogenesis) and functional recovery (skilled motor task). They also propose a molecular mechanism underlying the reparative effects of SVZ cells, which involves expression of VEGF.

The authors use markers such as Sox2 and Id2 (Fig1) to propose that the vast majority of SVZ-derived cells remain as progenitors even after infiltrating the cortex. However, these markers are also expressed in mature astrocytes. The authors should thus consider performing additional staining using these markers in combination with mature astrocyte markers to determine, among those Sox2+ (or Id2+), how many are in fact undifferentiated.

Even if many of the cells are at the progenitor state, it is unclear, with their experimental design, whether the SVZ-derived population actually remains in the undifferentiated state or whether there is continuous migration from the SVZ. What is the fate of these cells? Do they die after reaching the injured area? Or do they differentiate into glia?

Also, because they seem to record at each point cells at different maturational stages (e.g., activated NSCs, neuroblasts, mature glia and mature neurons), it is unclear at which point are the SVZ-derived cells beneficial to recovery (e.g., do they need to achieve certain level of differentiation in order to aid angiogenesis and synaptogenesis? Or are they more beneficial in the undifferentiated state?). The authors may consider depleting the SVZ-derived progeny at different stages of maturation to further investigate their observations.

It is proposed that cells are quiescent, possibly due to the author's quantification of Ki67+ cells, showing that only a minority of Tom+ cells are actively dividing. However, the authors show in figure 1M a region that has much fewer Tom+ cells than those displayed in panels G-M. If the authors quantified Ki67+ cells using a region more distant from the infarct area, it may have led to the conclusion that cells are mostly quiescent, because there may not be as many actively dividing cells further away from the lesion. The authors need to quantify the whole area, or consistently take areas at the same distance from the lesion for all their quantifications. Further, they may consider using a BrdU experiment to determine whether the cells are in fact quiescent throughout (or collecting samples at multiple times to get a more complete picture of the proliferative behavior).

The authors record a progressive increase in the number of SVZ-derived cells around the injury site. Is this due to proliferation in situ or due to protracted migration? At what point does this plateau after stroke and does functional recovery then follow the same trend?

The authors use GFAP-TK mice to chemogenetically ablate neural stem cells. Why wasn't the same Nestin promoter used in the lineage tracing also used in this context? Are the authors ablating the same population of cells that they saw migrating towards the injury site in the previous experiments? I.e., do Nestin+ and Gfap+ populations overlap? The authors could consider adding a staining (or references to previous published work) to validate that indeed the two markers can be used interchangeably to identify the same populations in the SVZ.

The authors refer to the SVZ-derived population as quiescent and undifferentiated. However, they also show that majority of these cells express Ascl1, which is a marker that has been identified in activated neural stem cells from the SVZ, and that is linked to the commitment of these cells to the neurogenic lineage fate. The authors may, thus, consider revising their nomenclature for these cells throughout the paper.

In Fig S4, the authors show no difference in Gfap signal in the peri-infarct area, between SVZ-depleted and control conditions. However, many of the depleted SVZ-derived cells should be generating Gfap+ cells. So, it is expected to have a reduction in the signal after GCV treatment. How do the authors explain the lack of differences between experimental conditions? Also, the authors should consider lineage tracing cells in the GCV condition to show that indeed the infiltration into the cortex is compromised.

Also, there is no mention of glial scar and other important processes happening during repair. Because many of the SVZ-derived cells are Gfap positive and develop into mature astrocytes, it is plausible that depleting the population would affect formation of the glial barrier between injured and intact tissue (and possibly also leading to changes in injury volume, which enigmatically the authors do not record). Have the authors looked into this?

It would be interesting to be able to dissect further the angiogenic and synaptogenic phenomena observed: does angiogenesis promote formation and survival of new spines through improved metabolic and trophic support or are the two directly influenced by VEGF expression?

The authors say in their discussion that there is no substantial cell replacement after stroke, and that the effects on repair they record are predominantly due to the presence of undifferentiated SVZ cells. However, they do record that around 30-40% of SVZ-derived cells mature into astrocytes (Fig1P). Thus, the study (along with other published work) suggests that there is, in fact, cell replacement, though this is predominantly limited to the glial compartment.

In the discussion (line 502), the authors say that it's commonly believed the main function of NSCs is to generate new neurons after injury. I think this is not accurate and I recommend rephrasing their

text: It is well established that neuronal cell replacement is poor, particularly in the cortex, and other studies have shown that NSCs migrating to cortical sites of damage tend to generate astrocytes rather than integrating into the circuit as new neurons. Rather than proposing that the main function of NSCs after stroke is that of generating new neurons, research focusing on cell replacement is rather interested in devising strategies to further promote the NSCs potential to generate new neurons.

Other concerns:

AAV5-Ef1a viruses will mostly infect neurons, but these are not expressing VEGF in normal conditions, according to the authors' quantifications. They may consider using different promoters to drive expression of VEGF in other cell types (e.g., Gfap?), so to avoid potential artefacts deriving from the ectopic expression in neurons.

In Fig2 and in the Methods section, the authors mention using several control conditions, but the plot only reports one column with controls. Do they find no differences between control groups? This should be mentioned somewhere, or the plots should display one column for each experimental condition.

In the drug administration section, what is the GCV concentration used?

The authors mention that in some experiment they changed the administration protocol and kept giving GCV after stroke. Which experiments are those and do the authors validate that there are no differences in the outcomes they record whether they only give GCV before the stroke vs giving it before and after?

In the behavioral testing, the authors say that they record which paw was preferentially used for reaching the pellet. Do they use this information for the injury model? E.g., do they injure the hemisphere linked to the preferred paw? Or do they control for this potential confounder in their behavioral analysis?

Reviewer #2 (Remarks to the Author):

Review of: Subventricular zone cytotgenesis provides trophic support for neural repair

FOR Nature Communications

In this article the authors are trying to ascertain how emigrated SVZ cells can be beneficial via neuro-replacement or neuroprotection. Williamson and colleagues show that SVZ precursor cells that have migrated to a model of stroke secrete VEGF, which is necessary for several post-stroke examples of plasticity. They also show that this process diminished with age and could be rescued by adding VEGF. This is translationally interesting and is consistent with the notion that neural precursor cells may help brain repair, not by replacing cells, but by secreting beneficial growth factors. They show that it is primarily precursors and not differentiated cells that migrate to photo-thrombotic cortical models of stroke.

The writing is lucid and the flow of the experiments logical. The images and quality of data is high and the depth of investigation appropriate (eg 100 cells examined per marker in Figure 1). The methods are clearly described. They appropriately control for changes in cortical lesion size. The photo-thrombotic lesion is a good choice in terms of controlling lesion placement and size.

A distinctive advantage of this paper is they show that reducing SVZ cytotgenesis in the stroke model diminishes recovery, especially behavioural (single seed reaching behaviour). This essential experiment has been remarkably under-utilised in the field. The 2-photon imaging of dendric spines after stroke combined with multi-exposure speckle blood flow imaging is a technical tour-de-force especially since they combine it with ganciclovir induced loss of cytotgenesis and behavioural assays. Importantly, they show that SVZ cells are needed for full spine plasticity.

They are correct to examine the SVZ response in aging as it diminishes dramatically in animal models but in humans is associated with more strokes. As expected, they show reduced SVZ proliferation after stroke in aging animals. This was associated with significantly worse performance in the seed reaching task and depleting SVZ precursors with ganciclovir did not make it worse. Finally, the viral VEGF delivery rescue experiment is fantastic. Behaviour, blood vessel growth and spine density are all rescued.

Overall, this is an interesting addition to the literature and methodologically it is quite strong. It is sure to generate follow-on studies showing how different growth factors may be secreted by SVZ cells in various models of neurological disease. I loved reading this paper but there are several aspects that could be strengthened.

Major points:

1. The idea that SVZ cells are beneficial via mechanisms other than cell replacement is not new. For example, neural stem cells from the SVZ have been shown to reduce inflammation and thereby be neuroprotective as the authors themselves acknowledge and cite 1. This should be better acknowledged.

2. The fact that it is primarily precursor cells that migrate towards the stroke does not mean that cell replacement does not occur. The precursors could gradually differentiate (perhaps even after 6 weeks post-injury) into more mature cells that do replace cells lost to injury. Also, the two events are not mutually exclusive. This should be acknowledged and in fact carrying out experiments on delayed differentiation would strengthen the paper, irrespective of what results they obtain.

3. It is interesting that loss of VEGF decreases both blood vessel growth and spine density. More discussion on the functional interactions of these two seemingly events is important to include. Do they regulate one another?

4. I also recommend justifying the choice of VEGF more thoroughly. Ultimately, would it not make sense that several trophic factors made by SVZ cells are at play? Please discuss.

5. Could the SVZ cells induce cortical cells to make and secrete growth factors? The staining for the latter seems to be in SVZ cells as well as in surrounding cortical cells.

6. It is the case that adult SVZ human neurogenesis is somewhat uncertain with the majority of immunohistochemistry studies supporting it but with a glaring paucity of other techniques available to either support it or refute it 2. This should be discussed in a more balanced fashion. It would be important to note that neonatal human SVZ neurogenesis is well accepted

7. The 6-week immunofluorescence data should be shown in the main section.

8. Adult SVZ stem and progenitor cells are not generally considered to be migratory, so the acquisition of this de novo phenotype after this model of stroke is interesting. How do you know it is not de-differentiation of adult neuroblasts?

Minor points:

9. In order to balance the discussion it may be useful to add a reference that suggests angiogenesis is not necessary for the repair process in stroke models 3.

10. Different models of stroke may elicit different responses. You should specify that in this model it is primarily precursors and astrocytes that migrate to the lesion but that in other models this may not be the case. I recommend you tone down the statement that this is the case across stroke models.

11. Is there any evidence that growth factors are secreted by SVZ cells that have emigrated to striatal MCAO strokes?

12. Fig. 5 – immunohistochemistry is notoriously difficult to quantify and this data would be strengthened by Western blots. Fig. 5D-G the turquoise font should be green.

13. “Neuroblasts originating in the SVZ migrate along vascular scaffolds towards the olfactory bulb in the healthy brain 50 and towards peri-infarct regions after stroke.

14. Do scRNAseq papers suggest that precursor cells preferentially make growth factors such as VEGF compared to more differentiated cells?

15. 12,51”. This aspect of SVZ migration is overstated and inadequately shown. Simply put there are blood vessels everywhere in the brain and it is inevitable that migrating neuroblasts contact them. But it does not mean that they need them for migration. Also, virtually nothing is known about the molecular regulation of ectopic emigration of stem / progenitor cells.

References

1 Pluchino, S. et al. Neurosphere-derived multipotent precursors promote neuroprotection by an immunomodulatory mechanism. *Nature* 436, 266-271 (2005).

2 Gault, N. & Szele, F. G. Immunohistochemical evidence for adult human neurogenesis in health and disease. *WIREs Mech Dis* 13, e1526, doi:10.1002/wsbm.1526 (2021).

3 Young, C. C. et al. Blocked angiogenesis in Galectin-3 null mice does not alter cellular and behavioral recovery after middle cerebral artery occlusion stroke. *Neurobiol Dis* 63, 155-164, doi:10.1016/j.nbd.2013.11.003 (2014).

Reviewer #3 (Remarks to the Author):

In this new manuscript, Williamson and colleagues assess the role played by neural precursor cells (that arise from the subventricular zone) in ischemic stroke outcomes in mice. They use elegant strategies to demonstrate that these migrating precursors migrate towards the peri-infarct region and produce growth factors (in particular VEGF) to promote recovery. The most interesting finding is that most of these migrating cells remain undifferentiated. Counterintuitively, they do not become mature neurons (that would integrate the network). Rather, these cells serve a trophic support for post-injury repair.

This manuscript is well written, the data of good quality and the figures well organized. Yet, while this study brings novel insight into cellular and molecular players in stroke recovery in mice, some concerns need to be addressed. Please see detailed comments below:

Major points:

- The most important question which requires clarification is what defines the path taken by precursor cells to migrate from the SVZ to the peri-infarct region. As authors mention, it is known that these cells migrate along blood vessels, but in this manuscript, it remains very descriptive. Yet, not characterizing the interaction between precursors and their direct environment is a missed opportunity. Is the path taken (which seems to delineate a trapeze-shaped region around the core) defined by remodeling of axons and/or dendrites? Or by the glial scar? Why would precursor cell migration be limited to that restricted region? Are guidance cues expressed nearby, at the borders? Etc. Answers to these questions would bring a lot more novelty to the study.
- Following up on the point above: in the context of this study, it is of utmost importance to further investigate the interaction between precursor cells and blood vessels, particularly as authors identify VEGF as key growth factor driving post-stroke repair. Describing the occurrence of physical interactions (Figure 5A-C) is not novel, unless a functional aspect is added to it. A major question that remains unanswered is whether perturbing vascular remodeling would impact migration patterns of precursor cells post-stroke. The authors might have an answer to this in their material. Indeed, they show that GCV treatment affects vascular remodeling in GFAP-TK mice. However, they do not show if the resulting reduction in vascular density affects migration patterns. Also, when conditional VEGF KO is performed, authors do not look at precursor migration patterns. This needs to be addressed, as new data could finally demonstrate a direct link between vascular architecture and precursor migration with a mechanistic value: Do lack of VEGF and perturbed vascular remodeling affect neural stem cell migration, and consequently worsen stroke outcomes?
- Results p.5 (related to Figure 1). Did lineage-traced cells become quiescent at this time point? It is worth checking at an earlier time point. What about one-week post-stroke? Authors could better cover the post-stroke plasticity period (from opening to closure).
- Results Figure 2L: authors must add/display untreated controls.
- Results Figure 2I: Does GCV itself affect the proliferation of endothelial cells? (if yes, it could impact reparative angiogenesis and ensuing migration of stem cells).

Minor points:

- Abstract: Try to better articulate the second half, as it is not very fluid. Also, it is suggested to replace “neural repair” by “neurovascular repair” (last sentence).
- Introduction: I am not sure if the very first statement is accurate (mouse vs. humans).
- Introduction p.4: typo “stroke in mice” (singular).
- Astrocytes are better defined by production of pan-astroglial marker ALDH1L1; it would be nice to have at least one image confirming their identity using this marker (since GFAP is also expressed by neural progenitors).
- Figure 3F: please indicate precise age on figure panels.
- Figure 7 title: please replace “due to” by “caused by”.

Responses to reviewer comments are indented and shown in blue.

Reviewer #1 (Remarks to the Author):

Williamson et al. present a study that focuses on the role of subventricular zone (SVZ)-derived cells in promoting recovery after stroke. They propose that cortical stroke triggers substantial migration of progenitors, and that these remain in an undifferentiated state long after injury has been inflicted. Using SVZ-cell ablation studies, they see a beneficial effect of these cells on tissue repair (in the form of synaptogenesis and angiogenesis) and functional recovery (skilled motor task). They also propose a molecular mechanism underlying the reparative effects of SVZ cells, which involves expression of VEGF.

The authors use markers such as Sox2 and Id2 (Fig1) to propose that the vast majority of SVZ-derived cells remain as progenitors even after infiltrating the cortex. However, these markers are also expressed in mature astrocytes. The authors should thus consider performing additional staining using these markers in combination with mature astrocyte markers to determine, among those Sox2+ (or Id2+), how many are in fact undifferentiated.

We used a wide variety of markers to explore the identity of lineage traced cells in Figure 1. We included Fig. 1E as a schematic to help with the interpretation of these data, but this comment prompted us to rearrange the figure and rewrite the associated Results section and legend to improve the clarity of our results. Note that we do not claim all Sox2+ or ID2+ cells to be progenitor cells. We found that >90% of cells expressed the classical neural precursor cell markers Sox2 and CD133. As the reviewer notes, and as we note in the Results (“Astrocyte reactivity is associated with re-expression of some precursor cell-associated proteins, including CD133 and Sox2^{24,25}”), these markers themselves do not necessarily demonstrate a precursor cell fate. We therefore used additional markers to further clarify the identity of these cells. Since reactive astrocytes do not express Ascl1 (refs 26, 27), and Ascl1 is a precursor cell marker, we used expression of Ascl1 to identify undifferentiated precursors. While many stem and progenitor cells also express common astrocyte markers (e.g., GFAP, Sox9, Aldh1l1), S100 β is a widely-expressed astrocyte marker that is not expressed by precursor cells. We have clarified (p. 5) that S100 β “has been shown to define astrocyte maturation and loss of multipotency^{21,22}”, and is therefore a suitable marker of mature, differentiated astrocytes. Thus, we used expression of Ascl1 and S100 β to identify undifferentiated precursors and differentiated astrocytes, respectively.

Even if many of the cells are at the progenitor state, it is unclear, with their experimental design, whether the SVZ-derived population actually remains in the undifferentiated state or whether there is continuous migration from the SVZ. What is the fate of these cells? Do they die after reaching the injured area? Or do they differentiate into glia?

Thank you for raising these points. To answer these questions we have included new data on lineage traced cells so that we now examine their identity at 1, 2, 6, and 8 weeks after stroke. While we observe increasing numbers of cells until 6 weeks, very few of the cells in peri-infarct cortex are proliferative at these times (Figure 1P, Q, shown below). Furthermore, we report expansion of the precursor cell pool and heightened proliferation within the SVZ after stroke (Fig. 3A-D). Together these data suggest that the increase in cell number over time is largely a result of continuous migration from the SVZ rather than local proliferation. Regarding the question of whether these cells differentiate, we now show that there is no difference in the distribution of cell identity across all of these survival times (1, 2, 6, and 8 weeks post-stroke; Fig. 1R, S, shown below), indicating that these cells do not differentiate over time even long after behavioral function has recovered. To examine the extent of cell death among the lineage traced population, we have added staining for the apoptotic marker cleaved caspase 3. We found that none of the >400 cells analyzed were cleaved caspase 3+ (Supplementary Fig. 1M, shown below). Combined with the increase in cell number over time, these data indicate rather little cell death among the lineage traced population.

Figure 1P-S:

Supplementary Fig. 1M:

Also, because they seem to record at each point cells at different maturational stages (e.g., activated NSCs, neuroblasts, mature glia and mature neurons), it is unclear at which point are the

SVZ-derived cells beneficial to recovery (e.g., do they need to achieve certain level of differentiation in order to aid angiogenesis and synaptogenesis? Or are they more beneficial in the undifferentiated state?). The authors may consider depleting the SVZ-derived progeny at different stages of maturation to further investigate their observations.

At all time points examined, the lineage traced cells had a similar distribution of phenotype (we now show time points from 1 to 8 weeks post-stroke in Fig. 1, see Fig. 1S in particular). Therefore, the timing of cell ablation would not answer the question of phenotype-function relationship since there would be no identity-selective ablation. This data also suggests that there may not be a need for these cells to differentiate in order to provide benefits for repair. In addition, the task of altering the timing of ablation is not possible with our system because the majority of lineage traced cells lose GFAP expression by 7 days so GCV administration would not kill them. In our study, we administered GCV prior to stroke so that the precursor cell population was depleted before the injury. While the idea of selectively ablating progeny of each identity is interesting, it would require the creation and validation of several new tools (one to allow the ablation of each cell type) that do not currently exist. This would be a very substantial undertaking that is best left for future studies. We have included the following in the Discussion (p. 18): “Additional work will be needed to clarify the mechanisms that dictate the identity of SVZ cells responding to stroke, and whether different cell types have distinct reparative functions.”

Also note that in Fig. 5J we report that VEGF is produced by all cell types except for neurons among the tdTomato⁺ cells. This finding suggests that all SVZ-derived cells except those with a neuronal identity may contribute to the VEGF-dependent mechanism we describe.

It is proposed that cells are quiescent, possibly due to the author’s quantification of Ki67⁺ cells, showing that only a minority of Tom⁺ cells are actively dividing. However, the authors show in figure 1M a region that has much fewer Tom⁺ cells than those displayed in panels G-M. If the authors quantified Ki67⁺ cells using a region more distant from the infarct area, it may have led to the conclusion that cells are mostly quiescent, because there may not be as many actively dividing cells further away from the lesion. The authors need to quantify the whole area, or consistently take areas at the same distance from the lesion for all their quantifications. Further, they may consider using a BrdU experiment to determine whether the cells are in fact quiescent throughout (or collecting samples at multiple times to get a more complete picture of the proliferative behavior).

We have clarified in the Methods that quantification for all markers, including Ki67, was done throughout peri-infarct cortex: “For immunohistochemical analysis of lineage traced cell identity, images were taken and cells were examined throughout peri-infarct cortex.” We show in Supplementary Figure 1L data examining the spatial distribution of lineage traced cells co-expressing various markers. We omitted Ki67 from this analysis because Ki67-expressing tdTomato⁺ cells were extremely rare across all regions of peri-infarct cortex. We also have added new data across 1 to 8 weeks after stroke (Fig. 1Q), which shows a similarly

low number of Ki67⁺ cells at all time points. Lastly, >90% of cells express the quiescence-associated marker Id2 (Fig. 1Q), which further strengthens our conclusion that the cells are largely quiescent.

The authors record a progressive increase in the number of SVZ-derived cells around the injury site. Is this due to proliferation in situ or due to protracted migration? At what point does this plateau after stroke and does functional recovery then follow the same trend?

These are important questions. Based on our newly added data, the number of SVZ-derived cells increased from 1 to 6 weeks post-stroke, which coincides with the period of functional improvement (Fig. 1P and 2L). Very few of the cells in peri-infarct cortex are Ki67⁺ across all of these times, suggesting limited in situ proliferation (Fig. 1Q). Furthermore, we report expansion of the precursor cell pool and heightened proliferation in the SVZ after stroke, which suggests continuous production of cells from the SVZ (Fig. 3A-D). Together these data suggest that the increase in cell number over time is largely a result of continuous migration from the SVZ rather than local proliferation. We have updated the text on page 6 to emphasize these points.

The authors use GFAP-TK mice to chemogenetically ablate neural stem cells. Why wasn't the same Nestin promoter used in the lineage tracing also used in this context? Are the authors ablating the same population of cells that they saw migrating towards the injury site in the previous experiments? I.e., do Nestin⁺ and Gfap⁺ populations overlap? The authors could consider adding a staining (or references to previous published work) to validate that indeed the two markers can be used interchangeably to identify the same populations in the SVZ. The authors refer to the SVZ-derived population as quiescent and undifferentiated. However, they also show that majority of these cells express *Ascl1*, which is a marker that has been identified in activated neural stem cells from the SVZ, and that is linked to the commitment of these cells to the neurogenic lineage fate. The authors may, thus, consider revising their nomenclature for these cells throughout the paper.

Thank you for raising this important point. We used these tools since we had established protocols for their use in our labs in previous studies. As suggested, we have added staining showing the overlap of Nestin⁺ and GFAP⁺ cells in the SVZ (Supplementary Figure 3, also shown below). We have also added citations (References 9, 10, 30, 31; PMID: 9185542, 12684469, 15494728, 10380923) that support that neural stem cells express both GFAP and Nestin. Furthermore, we verified that there is a loss of tdTomato⁺ Nestin-Cre^{ERT2} lineage traced cells when precursors are chemogenetically depleted (this is discussed more in our response to the next comment).

While *Ascl1* has been described as a “neurogenic” transcription factor, as the reviewer suggests, this notion appears to derive primarily from evidence that it 1) is implicated in neurogenesis, and 2) biases progeny to a neuronal fate when overexpressed far beyond normal biological levels. However, *Ascl1* is similarly required for developmental gliogenesis and lineage traced progeny from *Ascl1*⁺ precursors in the adult brain are multipotent, forming astrocytes, oligodendrocytes, and neurons (PMID: 25249462, 18032648; Supplementary Fig. 1). *Ascl1* is expressed in a population of multipotent progenitor cells that are generally regarded as immediately downstream of, or perhaps partially overlapping, *Nestin*⁺/*GFAP*⁺ neural stem cells (PMID: 31493429). Therefore, *Ascl1* is best defined as a marker of undifferentiated neural precursors that lie on the spectrum of differentiation between neural stem cells and differentiated neural cells. For these reasons, we refer to the *Ascl1*⁺ population of cells as undifferentiated precursors. This identity is also supported by the process of elimination in our data from Fig. 1R and S, since the proportion of cells that do not express markers of differentiated neural cells (astrocytes, oligodendrocytes, or neurons) aligns with the proportion of *Ascl1*-expressing cells. We also refer to the lineage traced cells as largely quiescent due to rare expression of the proliferation marker *Ki67* and near ubiquitous expression of the quiescence marker *Id2* (Fig. 1Q). We have revised the Results sections (p. 5-6) to clarify these points.

In Fig S4, the authors show no difference in *Gfap* signal in the peri-infarct area, between SVZ-depleted and control conditions. However, many of the depleted SVZ-derived cells should be generating *Gfap*⁺ cells. So, it is expected to have a reduction in the signal after GCV treatment. How do the authors explain the lack of differences between experimental conditions? Also, the authors should consider lineage tracing cells in the GCV condition to show that indeed the infiltration into the cortex is compromised.

While some of the SVZ-derived cells are *GFAP*⁺, we think the reason we did not detect a difference in the total *GFAP*⁺ glial scar between conditions is that the signal from SVZ-derived *GFAP*⁺ cells is small relative to that from parenchymal reactive astrocytes, which undergo substantial proliferation after injury to increase in number and very strongly upregulate *GFAP* expression. As suggested, we have added data as Supplementary Figure 3 (and shown below), which confirms the loss of lineage traced cells in the SVZ and migrating to cortex in *Ai14*; *NestinCre*^{ERT2}; *GFAP-TK* mice given GCV. We have added to the Results

(p. 6) that “GFAP-TK mice allow for ablation of the same population targeted by the Nestin-Cre^{ERT2} mice we used for lineage tracing (Supplementary Fig. 3)”

Also, there is no mention of glial scar and other important processes happening during repair. Because many of the SVZ-derived cells are Gfap positive and develop into mature astrocytes, it is plausible that depleting the population would affect formation of the glial barrier between injured and intact tissue (and possibly also leading to changes in injury volume, which enigmatically the authors do not record). Have the authors looked into this?

As discussed in response to the previous comment, we found no difference in the GFAP+ signal of the glial scar between mice with depleted precursors and controls (Supplementary Fig. 4). Regarding the point on injury volume, we did assess lesion volume in all experiments involving behavioral testing. We found no difference between groups in all cases. These data on lesion volume can be found in Figures 2M, 3L, 4D and Supplementary Figures 8 and 9.

It would be interesting to be able to dissect further the angiogenic and synaptogenic phenomena observed: does angiogenesis promote formation and survival of new spines through improved metabolic and trophic support or are the two directly influenced by VEGF expression?

This is an interesting question. There is evidence that VEGF can directly induce neuronal growth, as well as evidence that improved blood flow can influence synapse reorganization. We have revised the Discussion to better address this point (p. 17): “We identified VEGF produced by SVZ-derived cells as critical for effective repair and recovery after stroke. The potent angiogenic effects of VEGF have been well studied^{57,58}, but its effects on neuronal growth are less well understood. It is possible that VEGF produced by SVZ-derived cells directly enhanced neuronal outgrowth and synapse formation since ex vivo evidence suggests that VEGF increases neuronal complexity⁵⁶. An additional possibility is that restored blood flow due to VEGF-mediated angiogenesis provided metabolic support for neuronal growth^{7,64}. Indeed, growth of new blood vessels after stroke supports local blood flow increases³, which in turn contributes to the degree of local synaptogenesis⁵⁰.” Directly answering this question is best left for future studies as it is outside of the main point of our manuscript and would require the generation of multiple cell-type specific VEGF receptor

knockout mouse lines. It is also likely that both suggestions are true (direct and indirect actions of VEGF on neurons).

The authors say in their discussion that there is no substantial cell replacement after stroke, and that the effects on repair they record are predominantly due to the presence of undifferentiated SVZ cells. However, they do record that around 30-40% of SVZ-derived cells mature into astrocytes (Fig1P). Thus, the study (along with other published work) suggests that there is, in fact, cell replacement, though this is predominantly limited to the glial compartment.

We agree with this comment and have revised the text throughout the manuscript to reflect that there is some, mostly glial, cell replacement.

In the discussion (line 502), the authors say that it's commonly believed the main function of NSCs is to generate new neurons after injury. I think this is not accurate and I recommend rephrasing their text: It is well established that neuronal cell replacement is poor, particularly in the cortex, and other studies have shown that NSCs migrating to cortical sites of damage tend to generate astrocytes rather than integrating into the circuit as new neurons. Rather than proposing that the main function of NSCs after stroke is that of generating new neurons, research focusing on cell replacement is rather interested in devising strategies to further promote the NSCs potential to generate new neurons.

We have removed this text.

Other concerns:

AAV5-Ef1a viruses will mostly infect neurons, but these are not expressing VEGF in normal conditions, according to the authors' quantifications. They may consider using different promoters to drive expression of VEGF in other cell types (e.g., Gfap?), so to avoid potential artefacts deriving from the ectopic expression in neurons.

We deliberately chose to use the ubiquitous EF1 α promoter in order to induce VEGF expression across cell types since our goal was broadly to replace VEGF. However, the question of whether neuronal expression of VEGF directly contributed to some of the changes we observed (most notably the neuronal change of increased spine density) is an interesting one. Therefore, we created a GFAP-VEGF AAV to drive VEGF expression in astrocytes. We validated that this virus induced astrocytic VEGF expression without inducing neuronal expression. We examined spine density on apical dendrites of peri-infarct neurons and found that that GFAP-VEGF increased spine density relative to the control condition (CaMKII-eGFP alone). These data show that the effects of VEGF on synaptogenesis do not rely on VEGF expression in neurons. Whether the effects are due to direct action of VEGF on neurons or indirectly (e.g., through enhancement of angiogenesis and blood supply) remain to be answered, but will require substantial experimental effort that is outside the scope of our manuscript. We have included this new data as Supplementary Figure 11 (also shown below).

In Fig2 and in the Methods section, the authors mention using several control conditions, but the plot only reports one column with controls. Do they find no differences between control groups? This should be mentioned somewhere, or the plots should display one column for each experimental condition.

We have added new data as Figure 2L (also shown below) showing no difference in functional recovery between wildtype+saline, wildtype+GCV, and GFAP-TK+saline control conditions. We have added the following to the Methods (p. 20-21): “We found no differences in functional recovery between the following control groups: wildtype+saline, wildtype+GCV, GFAP-TK+saline (Fig. 2L). These conditions were combined into a single control group where exact conditions are not otherwise specified.”

In the drug administration section, what is the GCV concentration used?

We have clarified that the concentration was “typically 25-30 mg/mL, adjusted for pump rate.”

The authors mention that in some experiment they changed the administration protocol and kept giving GCV after stroke. Which experiments are those and do the authors validate that there are no differences in the outcomes they record whether they only give GCV before the stroke vs giving it before and after?

Thank you for bringing this point to our attention. The experiments from Figure 2 included administration before stroke; all others included before and again beginning two weeks after stroke (unless animals were euthanized prior to the two week point). We have depicted this on the experimental timelines for each figure involving GCV administration for clarification. We have also added to the Methods: “We did not observe any differences in animal behavior between the single and double dose regimens.”

In the behavioral testing, the authors say that they record which paw was preferentially used for reaching the pellet. Do they use this information for the injury model? E.g., do they injure the hemisphere linked to the preferred paw? Or do they control for this potential confounder in their behavioral analysis?

Thank you for pointing out this missing information. We have clarified that stroke was always induced in the hemisphere contralateral to the preferred paw.

Reviewer #2 (Remarks to the Author):

Review of: Subventricular zone cytotgenesis provides trophic support for neural repair

FOR Nature Communications

In this article the authors are trying to ascertain how emigrated SVZ cells can be beneficial via neuro-replacement or neuroprotection. Williamson and colleagues show that SVZ precursor cells that have migrated to a model of stroke secrete VEGF, which is necessary for several post-stroke examples of plasticity. They also show that this process diminished with age and could be rescued by adding VEGF. This is translationally interesting and is consistent with the notion that neural precursor cells may help brain repair, not by replacing cells, but by secreting beneficial growth factors. They show that it is primarily precursors and not differentiated cells that migrate to photo-thrombotic cortical models of stroke.

The writing is lucid and the flow of the experiments logical. The images and quality of data is high and the depth of investigation appropriate (eg 100 cells examined per marker in Figure 1). The methods are clearly described. They appropriately control for changes in cortical lesion size. The photo-thrombotic lesion is a good choice in terms of controlling lesion placement and size. A distinctive advantage of this paper is they show that reducing SVZ cytotgenesis in the stroke model diminishes recovery, especially behavioural (single seed reaching behaviour). This essential experiment has been remarkably under-utilised in the field. The 2-photon imaging of dendric spines after stroke combined with multi-exposure speckle blood flow imaging is a technical tour-de-force especially since they combine it with ganciclovir induced loss of cytotgenesis and behavioural assays. Importantly, they show that SVZ cells are needed for full spine plasticity.

They are correct to examine the SVZ response in aging as it diminishes dramatically in animal models but in humans is associated with more strokes. As expected, they show reduced SVZ proliferation after stroke in aging animals. This was associated with significantly worse performance in the seed reaching task and depleting SVZ precursors with ganciclovir did not make it worse. Finally, the viral VEGF delivery rescue experiment is fantastic. Behaviour, blood vessel growth and spine density are all rescued.

Overall, this is an interesting addition to the literature and methodologically it is quite strong. It is sure to generate follow-on studies showing how different growth factors may be secreted by SVZ cells in various models of neurological disease. I loved reading this paper but there are several aspects that could be strengthened.

Major points:

1. The idea that SVZ cells are beneficial via mechanisms other than cell replacement is not new. For example, neural stem cells from the SVZ have been shown to reduce inflammation and thereby be neuroprotective as the authors themselves acknowledge and cite 1. This should be better acknowledged.

We agree and have added additional mention of these studies throughout the manuscript.

P. 9: “several studies have demonstrated beneficial effects of transplanted stem cells...”

P. 10: “... neural precursors of various sources and in diverse disease settings have been reported to express trophic factors, which may be implicated in their therapeutic effects 40,41,44–46,51–53 ”

P. 19: “... there is evidence that precursor cells can attenuate inflammation and have protective effects after injury...”

2. The fact that it is primarily precursor cells that migrate towards the stroke does not mean that cell replacement does not occur. The precursors could gradually differentiate (perhaps even after 6 weeks post-injury) into more mature cells that do replace cells lost to injury. Also, the two events are not mutually exclusive. This should be acknowledged and in fact carrying out experiments on delayed differentiation would strengthen the paper, irrespective of what results they obtain.

Thank you for raising this important point. We have made changes to address this comment in two ways. First, we agree that our data does not support an absence of cell replacement and have revised text throughout the manuscript to reflect this. Second, we have added new data from additional time points. We now include data from 1, 2, 6, and 8 weeks after injury (Figure 1; Fig. 1R and S are shown below). We find that across all timepoints the identity of SVZ-derived cells is very similar, with the majority being precursors. These data suggest that there is not delayed differentiation over this time period. It is possible that at even later survival times there is delayed differentiation. We have added to the discussion (p. 17): “We did not find evidence of delayed differentiation of lineage traced cells out to 8 weeks post-stroke, which is well after the period during which substantial functional improvement occurs. It remains possible that there is some delayed differentiation of these cells at even later time points.”

3. It is interesting that loss of VEGF decreases both blood vessel growth and spine density. More discussion on the functional interactions of these two seemingly events is important to include. Do they regulate one another?

This is an interesting point. We have revised the Discussion to better address this question (p. 17-18): “We identified VEGF produced by SVZ-derived cells as critical for effective repair and recovery after stroke. The potent angiogenic effects of VEGF have been well

studied^{57,58}, but its effects on neuronal growth are less well understood. It is possible that VEGF produced by SVZ-derived cells directly enhanced neuronal outgrowth and synapse formation since ex vivo evidence suggests that VEGF increases neuronal complexity⁵⁶. An additional possibility is that restored blood flow due to VEGF-mediated angiogenesis provided metabolic support for neuronal growth^{7,71}. Indeed, growth of new blood vessels after stroke supports local blood flow increases³, which in turn contributes to the degree of local synaptogenesis⁵⁰.”

4. I also recommend justifying the choice of VEGF more thoroughly. Ultimately, would it not make sense that several trophic factors made by SVZ cells are at play? Please discuss.

We have clarified our motivation for focusing on VEGF based on our findings in Figure 5I: “Since VEGF was uniquely highly expressed in SVZ-derived cells relative to resident cortical cells, we next investigated whether VEGF produced by cells arising from the SVZ was involved in post-stroke recovery and repair” (p. 11). We agree that other factors may be involved and have added to the Discussion (p. 16-17): “Importantly, these past studies have identified numerous factors produced by precursor cells depending on context. It is possible that multiple factors produced by SVZ-derived cells promote recovery after stroke. This is suggested by our finding that recovery is worse in mice with ablated neural stem cells compared with VEGF cKO mice. Thus, future studies could examine other molecular targets.”

5. Could the SVZ cells induce cortical cells to make and secrete growth factors? The staining for the latter seems to be in SVZ cells as well as in surrounding cortical cells.

This is an interesting point. While other cortical cells express growth factors, expression (at least of the proteins we examined: BDNF, GDNF, FGF2) does not appear to be induced by SVZ-derived cells because it is present and unchanged in mice with ablated neural stem cells (Figure 5I). We have added to the Discussion (p. 17) that: “It is also conceivable that SVZ-derived cells could induce the expression of growth promoting factors in resident cortical cells, but this remains to be determined.”

6. It is the case that adult SVZ human neurogenesis is somewhat uncertain with the majority of immunohistochemistry studies supporting it but with a glaring paucity of other techniques available to either support it or refute it 2. This should be discussed in a more balanced fashion. It would be important to note that neonatal human SVZ neurogenesis is well accepted

As suggested, we have expanded our Discussion on this topic, including by citing Gault and Szele (p. 19): “The persistence of neural stem cells and cytogenesis in adult humans is debated. While it is generally accepted to occur perinatally and in children, there is evidence for and against SVZ cytogenesis in the healthy and injured adult human brain⁶⁴⁻⁷⁰. If

cytogenesis declines in aging humans, as in rodents, our study provides rationale for several treatment strategies: slowing neural stem cell decline, enhancing neural stem cell activation, and replacing factors produced by the neural stem cell lineage.”

7. The 6-week immunofluorescence data should be shown in the main section.

We now include this data in Figure 1 along with newly collected data from 1 and 8 weeks post-stroke.

8. Adult SVZ stem and progenitor cells are not generally considered to be migratory, so the acquisition of this de novo phenotype after this model of stroke is interesting. How do you know it is not de-differentiation of adult neuroblasts?

There are five reasons why we think the SVZ-derived cells that localize in peri-infarct cortex are unlikely to arise from de-differentiated neuroblasts. First, we are unaware of any evidence obtained using stringent lineage tracing techniques that suggests neuroblasts are capable of de-differentiation in vivo. Second, the population of DCX+ lineage traced cells in peri-infarct cortex is comparably small (around 1-2%) across all timepoints assessed (1 to 8 weeks post-stroke). We would expect to see a considerably larger proportion of DCX+ cells, especially early on, if it was the case that neuroblasts migrated to cortex and then de-differentiated. Third, we found no evidence for any changes in cell identity with distance from the infarct (Supplementary Figure 1L). DCX+ cells are similarly rare among the lineage traced cells near the SVZ that have begun to migrate to cortex. We would expect cells far from the infarct to be DCX+ if the migrating cells were neuroblasts that then changed identity. These observations indicate that the migratory cells are very rarely neuroblasts. Fourth, transplantation studies have shown that transplanted neural stem cells exhibit directed migration towards sites of injury (e.g., PMID: 27733606, 15608062, 19617198), which provides evidence that neural stem cells do have migratory potential. Fifth, newly added data (Fig. 8) shows that CXCL12-CXCR4 interactions contribute to post-injury migration of the lineage traced population. Since CXCR4 expression is common to all lineage traced cells arising from the SVZ (Fig. 8D), cell types other than neuroblasts express the machinery for directed migration, at least in the context of this post-stroke migratory response.

Minor points:

9. In order to balance the discussion it may be useful to add a reference that suggests angiogenesis is not necessary for the repair process in stroke models 3.

We now cite Young et al. for balance, as suggested.

10. Different models of stroke may elicit different responses. You should specify that in this

model it is primarily precursors and astrocytes that migrate to the lesion but that in other models this may not be the case. I recommend you tone down the statement that this is the case across stroke models.

As suggested, we have clarified that our conclusions about cell identity come from the photothrombosis model: “These experiments identify undifferentiated precursors as the predominant cell type produced by the SVZ in response to photothrombotic stroke.”

11. Is there any evidence that growth factors are secreted by SVZ cells that have emigrated to striatal MCAO strokes?

Only a very small number of studies have lineage traced SVZ-derived cells after MCAO, and none have examined growth factor expression. We have added to the discussion that “It is also possible that the nature of the SVZ migratory response and the factors produced by these cells might be different in other stroke models.” (p. 18).

12. Fig. 5 – immunohistochemistry is notoriously difficult to quantify and this data would be strengthened by Western blots. Fig. 5D-G the turquoise font should be green.

Quantification of protein expression by immunofluorescence has become widely-adopted and reliable due to advances in imaging and fluorescent probe technologies. Intensity-based quantification of immunofluorescence images provides useful data when staining, imaging, and analysis parameters are optimized and uniformly applied to all samples (PMID: 16978205, 33553685, 37368874), as we have done. Indeed, it has been shown that there is a strong concordance between immunofluorescent intensity-based and mass spectrometry-based determination of protein content (e.g., PMID: 28092364). The immunohistochemistry approach is particularly preferable to western blot in scenarios such as ours where small regions of interest (i.e. peri-infarct cortex) border others, such as the infarct, that would be expected to have very different abundance of the protein of interest, and could thus be highly influenced by the precision of tissue dissection. Finally, our conditional KO experiments provide the gold-standard evidence that VEGF is produced by SVZ-derived cells and is important for repair and recovery after stroke.

We have corrected the font color, as suggested.

13. “Neuroblasts originating in the SVZ migrate along vascular scaffolds towards the olfactory bulb in the healthy brain 50 and towards peri-infarct regions after stroke. 12,51”. This aspect of SVZ migration is overstated and inadequately shown. Simply put there are blood vessels everywhere in the brain and it is inevitable that migrating neuroblasts contact them. But it does not mean that they need them for migration. Also, virtually nothing is known about the molecular regulation of ectopic emigration of stem / progenitor cells.

We have removed the sentence in question.

14. Do scRNAseq papers suggest that precursor cells preferentially make growth factors such as VEGF compared to more differentiated cells?

To our knowledge, no scRNAseq data includes lineage traced SVZ-derived cells after stroke, which would be desired to either purify or informatically isolate this cell population since it is relatively small and its transcriptional profile is not well defined. It would also be necessary to ensure that tissue was dissected so as to separate the peri-infarct region from the SVZ since SVZ cells could be classified similarly to migrating cells due to overlap of some markers. This experiment has not yet been done.

References

- 1 Pluchino, S. et al. Neurosphere-derived multipotent precursors promote neuroprotection by an immunomodulatory mechanism. *Nature* 436, 266-271 (2005).
- 2 Gault, N. & Szele, F. G. Immunohistochemical evidence for adult human neurogenesis in health and disease. *WIREs Mech Dis* 13, e1526, doi:10.1002/wsbm.1526 (2021).
- 3 Young, C. C. et al. Blocked angiogenesis in Galectin-3 null mice does not alter cellular and behavioral recovery after middle cerebral artery occlusion stroke. *Neurobiol Dis* 63, 155-164, doi:10.1016/j.nbd.2013.11.003 (2014).

Reviewer #3 (Remarks to the Author):

In this new manuscript, Williamson and colleagues assess the role played by neural precursor cells (that arise from the subventricular zone) in ischemic stroke outcomes in mice. They use elegant strategies to demonstrate that these migrating precursors migrate towards the peri-infarct region and produce growth factors (in particular VEGF) to promote recovery. The most interesting finding is that most of these migrating cells remain undifferentiated.

Counterintuitively, they do not become mature neurons (that would integrate the network).

Rather, these cells serve a trophic support for post-injury repair.

This manuscript is well written, the data of good quality and the figures well organized. Yet, while this study brings novel insight into cellular and molecular players in stroke recovery in mice, some concerns need to be addressed. Please see detailed comments below:

Major points:

- The most important question which requires clarification is what defines the path taken by precursor cells to migrate from the SVZ to the peri-infarct region. As authors mention, it is known that these cells migrate along blood vessels, but in this manuscript, it remains very descriptive. Yet, not characterizing the interaction between precursors and their direct

environment is a missed opportunity. Is the path taken (which seems to delineate a trapeze-shaped region around the core) defined by remodeling of axons and/or dendrites? Or by the glial scar? Why would precursor cell migration be limited to that restricted region? Are guidance cues expressed nearby, at the borders? Etc. Answers to these questions would bring a lot more novelty to the study.

These questions are certainly intriguing and we have spent considerable time studying them. We wish to thank the reviewer for this comment because we believe the new data we have added to address it adds significant value to our manuscript. We hypothesized that blood vessels surrounding the infarct may be key drivers of SVZ cell migration given the close interactions between these cell types. We used recent molecular profiling data of vasculature after stroke (PMID: 37058487) to identify candidate mechanisms. In the newly added Figure 8 (shown below), we provide evidence that expression of the chemokine CXCL12 is markedly enhanced in vasculature surrounding the infarct, that migrating cells from the SVZ express the cognate receptor CXCR4, and that antagonizing CXCL12-CXCR4 interactions impairs the migratory response. We think that the chemoattractive CXCL12 gradient defines the migratory path, which is essentially a direct path from the dorsal SVZ towards the infarct. This would explain why the cells migrate towards the near aspect of the infarct but infrequently localize in very superficial regions. More detail on this new data can be found in the Results section (p. 14-15) and Figure 8.

- Following up on the point above: in the context of this study, it is of utmost importance to further investigate the interaction between precursor cells and blood vessels, particularly as authors identify VEGF as key growth factor driving post-stroke repair. Describing the occurrence of physical interactions (Figure 5A-C) is not novel, unless a functional aspect is added to it. A major question that remains unanswered is whether perturbing vascular remodeling would impact migration patterns of precursor cells post-stroke. The authors might have an answer to this in their material. Indeed, they show that GCV treatment affects vascular remodeling in GFAP-TK mice. However, they do not show if the resulting reduction in vascular density affects migration patterns. Also, when conditional VEGF KO is performed, authors do not look at

precursor migration patterns. This needs to be addressed, as new data could finally demonstrate a direct link between vascular architecture and precursor migration with a mechanistic value: Do lack of VEGF and perturbed vascular remodeling affect neural stem cell migration, and consequently worsen stroke outcomes?

We appreciate the reviewer's interest in this question and we actually included the experiment they describe in our original submission. While we cannot assess migration in GFAP-TK mice because the cells of interest are ablated, this is possible for the VEGF cKO manipulation as the reviewer suggests. For Figure 6D and E (shown below), we generated Nestin-CreER; Ai14; VEGF^{f/f} mice to test whether the number of SVZ-derived cells that migrate to peri-infarct cortex was reduced by conditional VEGF knockout, which resulted in impaired vascular remodeling and recovery. We found that the migration of cells to peri-infarct cortex in VEGF cKO mice was similar to those with intact VEGF, which suggests that the remodeling of vessels is not critical for the migration of cells from the SVZ.

This comment made us realize we did not adequately emphasize the importance of these findings. We have added the following (p. 12): “

“The finding that the number of lineage traced cells that localized to peri-infarct cortex was unaffected by VEGF cKO suggests that vascular remodeling is not required for the migration of cells from the SVZ.”

And on p. 15:

“Based on our findings, we propose the following model: vasculature surrounding the infarct produces guidance cues, such as CXCL12, that directs the ectopic migration of cells from the SVZ. In turn, these migrating cells produce VEGF to drive repair processes in the residual tissue surrounding the infarct.”

In this model, the remodeling of vasculature is not necessary for the migration of cells from the SVZ, but rather a beneficial consequence of their production of VEGF. The close interactions between SVZ-derived cells and vessels are also explained by the migrating cells following a chemokine gradient.

- Results p.5 (related to Figure 1). Did lineage-traced cells become quiescent at this time point? It is worth checking at an earlier time point. What about one-week post-stroke? Authors could better cover the post-stroke plasticity period (from opening to closure).

We have added data from additional time points, which now include 1, 2, 6, and 8 weeks post-stroke. We have included this data in a newly reorganized Figure 1. The data most relevant for this question about quiescence are shown below (Fig. 1Q). We find that at all time points examined, the lineage traced cells localized near the infarct are very rarely Ki67+, suggesting that they are quiescent throughout and after the recovery period. We have revised the text on pages 5 and 6 to emphasize these points.

- Results Figure 2L: authors must add/display untreated controls.

As suggested, we have added data from saline-treated controls (both wildtype and GFAP-TK mice). Both of these groups showed recovery similar to wildtype+GCV mice and significantly better than GFAP-TK+GCV mice.

- Results Figure 2I: Does GCV itself affect the proliferation of endothelial cells? (if yes, it could impact reparative angiogenesis and ensuing migration of stem cells).

We have added new data as Supplementary Figure 6 (also shown below) that shows no effect of GCV vs saline administration in wildtype mice on peri-infarct vascular density. This data indicates that GCV alone does not affect vascular remodeling after stroke.

Minor points:

- Abstract: Try to better articulate the second half, as it is not very fluid. Also, it is suggested to replace “neural repair” by “neurovascular repair” (last sentence).

We have rewritten the abstract. We decided to keep “neural repair” because it refers generally to nervous system tissue whereas “neurovascular” can be used to refer specifically to vasculature.

- Introduction: I am not sure if the very first statement is accurate (mouse vs. humans).

We have updated the sentence to: “Functional recovery is often limited after damage to the central nervous system.”

- Introduction p.4: typo “stroke in mice” (singular).

Thank you – fixed.

- Astrocytes are better defined by production of pan-astroglial marker ALDH1L1; it would be nice to have at least one image confirming their identity using this marker (since GFAP is also expressed by neural progenitors).

While ALDH1L1 is a great astrocyte marker, it is also expressed by neural precursor populations in the SVZ (PMID: 23836537, 34557065). It is therefore not the best marker to determine mature astrocyte identity in the context of this study since the identity of ALDH1L1⁺ cells would be ambiguous. This is also true of GFAP and Sox9, but not of S100 β (PMID: 28336567, 17078026). We therefore used expression of S100 β , which is a widely-expressed mature astrocyte marker, to confirm the identity of mature astrocytes and have clarified in the text that “S100 β has been shown to define astrocyte maturation and loss of multipotency^{21,22}, and is not expressed by SVZ precursors²³.”

- Figure 3F: please indicate precise age on figure panels.

Done.

- Figure 7 title: please replace “due to” by “caused by”.

Done.

REVIEWERS' COMMENTS

Reviewer #1 (Remarks to the Author):

The manuscript is of good quality, with sound methods and analysis, as well as novel and interesting results.

The authors have addressed concerns raised by the reviewers with clarifications of the work originally presented or inclusion of additional experiments supporting their conclusions.

I think the manuscript is suitable for publication in Nat Comm.

Reviewer #2 (Remarks to the Author):

Re-Review of: Subventricular zone cytotogenesis provides trophic support for neural repair

In general, the authors have done an excellent job in responding to my comments and suggestions which I list as appropriate responses. The paper now is much improved and now ready for publication in Nat. Comm. However before doing so, please address the following minor concerns.

Appropriate responses

1. Comment 2. Thank for adding the extra time points, especially the 8 week one and for addressing the possibility of delayed differentiation in the text.
2. Comment 3. The new discussion on VEGF and neuronal growth and synaptogenesis is a fine addition.
3. Comment 4. The authors discuss how factors other than VEGF could be at play.
4. Comment 5. As suggested, they also allude to the possibility of cortical cells expressing growth factors in addition to the SVZ.
5. Comment 6: The current controversy of adult human SVZ neurogenesis is well written.
6. Comment 7: The 6-week time point and the additional time points added to Fig. 1 are a good change.
7. Minor Comments 9 - 14 are all appropriately addressed.

Minor concerns.

8. Comment 1. Thank you for providing three citations on the beneficial roles of SVZ cells. However, I apologise if I was not clear. The point was that these beneficial effects can occur in the absence of neuroreplacement.
9. Comment 8. The five reasons the authors give for why they do not think emigrated Dcx+ cells de-differentiate and take on stem progenitor cell phenotype are well argued. I recommend adding a distillation of these arguments into the discussion. I also recommend including in the discussion that studies showing SVZ stem or progenitor migration to injuries are still comparatively rare compared to Dcx+ cell emigration. This point and the idea that stroke induced de novo stem and progenitor migration should be included.

Reviewer #3 (Remarks to the Author):

The revised version adequately addresses all points that I raised before. I wish to congratulate the authors for their excellent work.

Author comments are indented and shown in blue.

Reviewer #1 (Remarks to the Author):

The manuscript is of good quality, with sounds methods and analysis, as well as novel and interesting results.

The authors have addressed concerns raised by the reviewers with clarifications of the work originally presented or inclusion of additional experiments supporting their conclusions.

I think the manuscript is suitable for publication in Nat Comm.

We wish to thank the reviewer for their constructive feedback.

Reviewer #2 (Remarks to the Author):

Re-Review of: Subventricular zone cytogenesis provides trophic support for neural repair

In general, the authors have done an excellent job in responding to my comments and suggestions which I list as appropriate responses. The paper now is much improved and now ready for publication in Nat. Comm. However before doing so, please address the following minor concerns.

We wish to thank the reviewer for their constructive feedback.

Appropriate responses

1. Comment 2. Thank for adding the extra time points, especially the 8 week one and for addressing the possibility of delayed differentiation in the text.
2. Comment 3. The new discussion on VEGF and neuronal growth and synaptogenesis is a fine addition.
3. Comment 4. The authors discuss how factors other than VEGF could be at play.

4. Comment 5. As suggested, they also allude to the possibility of cortical cells expressing growth factors in addition to the SVZ.
5. Comment 6: The current controversy of adult human SVZ neurogenesis is well written.
6. Comment 7: The 6-week time point and the additional time points added to Fig. 1 are a good change.
7. Minor Comments 9 - 14 are all appropriately addressed.

Minor concerns.

8. Comment 1. Thank you for providing three citations on the beneficial roles of SVZ cells. However, I apologise if I was not clear. The point was that these beneficial effects can occur in the absence of neuroreplacement.

We have clarified this point (p. 11) “...which may be implicated in their therapeutic effects and may be independent of cell replacement”.

9. Comment 8. The five reasons the authors give for why they do not think emigrated Dcx+ cells de-differentiate and take on stem progenitor cell phenotype are well argued. I recommend adding a distillation of these arguments into the discussion. I also recommend including in the discussion that studies showing SVZ stem or progenitor migration to injuries are still comparatively rare compared to Dcx+ cell emigration. This point and the idea that stroke induced de novo stem and progenitor migration should be included.

We have added the following to the discussion (p. 17): “While neuroblast migration in the adult brain is well documented, migration of neural stem cells and undifferentiated precursors is rare^{12,20}. However, it is unlikely that migratory neuroblasts are the source of SVZ-derived cells that migrate to peri-infarct cortex. For instance, we found few neuroblasts along the migration route from the SVZ to the infarct across multiple time points after stroke. The de novo migration of precursor cells appears to be due to the unique and high expression of CXCL12 in peri-infarct regions.”

Reviewer #3 (Remarks to the Author):

The revised version adequately addresses all points that I raised before. I wish to congratulate the authors for their excellent work.

We wish to thank the reviewer for their constructive feedback.